# DIFFUSION-ENHANCED GFLOWNET FOR SOLVING VEHICLE ROUTING PROBLEMS

## ABSTRACT

Traditional neural solvers for solving vehicle routing problems (VRPs) often suffer from limited solution diversity, motivating the development of Generative Flow Network (GFlowNet)–based models. However, the effectiveness of these models is frequently constrained by insufficient flow expansion in high-reward regions, limiting their ability to distribute probability across promising solution routes, the deeper exploration could yield superior results. Diffusion models, in contrast, provide stronger structural guidance for exploration. These two paradigms are naturally complementary: GFlowNet can supply edge-level signals for diffusion to embed, while diffusion can guide broader exploration. Leveraging this synergy, we propose Diffusion-Enhanced GFlowNet (DEG), a novel framework that integrates GFlowNet with diffusion model to encourage richer flow expansion toward high-reward regions and derive higher-quality solutions. Specifically, DEG exploits GFlowNet's inherent diversity to generate edge-specific backward signals, applies the stochastic noise schedule of diffusion to perturb these signals, and then denoises them within the GFlowNet paradigm. To further improve scalability, we introduce a specialized decoder capable of dynamically adapting to diverse problem scales. Extensive experimental evaluations on synthetic and real-world datasets, including instances with up to 10,000 nodes, demonstrate that DEG consistently achieves favorable performance compared to baseline methods.

## 1 INTRODUCTION

The Vehicle Routing Problem (VRP) is a fundamental NP-hard combinatorial optimization problem that generalizes the Traveling Salesman Problem (TSP) by incorporating practical constraints such as vehicle capacity limitations and route feasibility requirements. The objective of VRPs is to minimize overall travel costs by determining optimal routes for a fleet of vehicles dispatched from a central depot to serve multiple customers while satisfying these constraints. VRPs are central to a wide range of applications, including logistics and distribution planning (Bochtis & Sørensen, 2009; 2010; Cattaruzza et al., 2017; Hsiao et al., 2018; Konstantakopoulos et al., 2022; Leng & Li, 2021), agricultural field operations (Wu et al., 2023; Yao et al., 2022; Utamima & Djunaidy, 2022), and urban delivery systems (Carrabs et al., 2017; Gayialis et al., 2019; Ćirović et al., 2014). Owing to their importance, VRPs have attracted extensive research attention. Traditional metaheuristics (Bräysy & Gendreau, 2001; Baker & Ayechew, 2003; Mohammed et al., 2017; Osman, 1993; Chiang & Russell, 1996; Kuo, 2010; Rizzoli et al., 2007; Montemanni et al., 2005; Bell & McMullen, 2004) have demonstrated effectiveness in solving VRPs. However, they are often computationally expensive and heavily dependent on carefully designed solution components and extensive hyperparameter tuning tailored to specific instances. These limitations have spurred growing interest in deep learning–based neural solvers, which aim to provide data-driven heuristics that improve inference efficiency while reducing reliance on expert knowledge.

Recent neural solvers for VRPs can be broadly categorized into Autoregressive (AR) and Non-Autoregressive (NAR) models. AR models, such as those based on pointer networks and Transformer architectures (Kwon et al., 2020; Luo et al., 2023; Bresson & Laurent, 2021), have achieved strong solution quality. However, their sequential decoding process introduces significant computational bottlenecks, limiting scalability to large instances. NAR models provide a promising alternative by generating complete solutions in a single forward pass, greatly improving inference

efficiency. Early NAR methods, however, struggled with solution quality, motivating the development of Generative Flow Network (GFlowNet) (Bengio et al., 2021; 2023). While GFlowNet-based solvers (Zhang et al., 2025; Kim et al., 2025) leverage sampling diversity to improve solution quality, they often converge to sub-optimal route subsets and fail to fully explore high-reward regions, leading to insufficient flow expansion into these more promising areas and hindering the discovery of higher-quality solutions.

Diffusion models (Sohl-Dickstein et al., 2015; Ho et al., 2020) recently emerged as a powerful approach in the VRP domain for their strong exploration capability (Sun & Yang, 2023; Li et al., 2023; 2024; Wang et al., 2025). Yet existing diffusion-based methods remain largely confined to the TSP and depend on supervised frameworks to generate edge-level noise signals. While GFlowNet and diffusion models each have distinct strengths and limitations, their integration for VRP solving has not been explored. These two paradigms are naturally complementary: diffusion can guide GFlowNet toward more effective flow expansion in high-reward regions, and GFlowNet can provide edge-level signals and extend the applicability of diffusion methods.

Based on this, we propose Diffusion-Enhanced GFlowNet (DEG), a novel framework that integrates GFlowNet with diffusion models for solving VRPs. DEG leverages the intrinsic diversity of GFlowNet to generate edge-specific backward signals, which serve as noise basis for the diffusion process. Unlike conventional trajectory-level backward signals that collapse into a single aggregated value, edge-specific signals provide richer structure for diffusion noise modeling. DEG perturbs these signals with a stochastic noise schedule and denoises them within the GFlowNet paradigm, preventing premature convergence to sub-optimal policies and promoting broader exploration of high-reward regions. This mechanism enhances forward probability allocation across promising routes and enables more effective flow expansion, enabling the discovery of higher-quality solutions. To further enhance generalization across varying problem scales, we develop the Graph-Scale Adapter (GSA), a specialized module that aligns spatial statistics across different graph scales and significantly improves scalability. Experimental evaluations demonstrate DEG's strong performance and scalability on both TSP and Capacitated Vehicle Routing Problem (CVRP), achieving competitive results compared to existing baselines and exhibiting robustness on instances containing up to 10,000 nodes. Our contributions are summarized as follows:

- We propose Diffusion-Enhanced GFlowNet (DEG), which leverages intrinsic diversity of GFlowNet to generate edge-specific backward signals that serve as the adaptive noise basis for integrating diffusion, then applies the diffusion's noise schedule to inject noise into these signals, and denoises them through the GFlowNet paradigm, enabling deeper flow expansion toward high-reward regions to derive better solution.

- We design the Graph-Scale Adapter (GSA) to align spatial statistics across varying instance sizes, substantially enhancing the model's ability to generalize to large-scale problems.

- We conduct comprehensive experiments on TSP and CVRP, and results show that DEG consistently achieves favorable results on both synthetic and real-world datasets.

## 2  RELATED WORK

The deep learning based neural solvers for VRPs can be broadly classified into Autoregressive (AR) and Non-Autoregressive (NAR) methods, respectively.

**AR methods** sequentially construct solutions step-by-step, conditioned on previously selected nodes. Transformer-based architectures have been particularly influential due to their ability to effectively capture long-range dependencies. A notable early example is the Attention Model (AM) (Kool et al., 2019), which applies self-attention mechanisms to enable end-to-end routing policy learning. To further enhance solution diversity and robustness, POMO (Kwon et al., 2020) builds on AM by employing multi-start policy optimization, exploring multiple potential optimal paths simultaneously. Building on the strengths of autoregressive architectures, LEHD (Luo et al., 2023) introduces a light-encoder, heavy-decoder design, achieving substantial performance gains across standard VRP benchmarks. More recently, L2R (Zhou et al., 2025) proposes a reduction-based candidate selection mechanism, which dynamically filters feasible actions based on the current state to further improve inference efficiency.

**NAR methods**, by contrast, generate complete solutions in a single forward pass, drastically reducing inference time and enabling parallel solution generation. This characteristic makes them particularly appealing for large-scale or real-time applications. Diffusion-based models represent a recent and promising class of NAR approaches. DIFUSCO (Sun & Yang, 2023) is pioneering diffusion-based model tailored for the TSP variant of VRPs. Building upon this foundation, T2T (Li et al., 2023), Fast T2T (Li et al., 2024) and DEITSP (Wang et al., 2025) significantly improve solution quality while retaining inference efficiency. However, all these diffusion-based methods rely on supervised learning with optimal solutions and are mainly limited to TSP. GFlowNet-based solvers form another mainstream approach. GFACS (Kim et al., 2025) employs GFlowNet to enhance the ACO heuristic and applies energy reshaping to refine the reward. AGFN (Zhang et al., 2025) adopts an adversarial strategy to generate routes in an end-to-end manner. But both methods are hindered by insufficient exploration of high-reward regions. To address this, we propose Diffusion-Enhanced GFlowNet(DEG), a generative framework that encourages flow expansion in high-reward regions.

## 3 PRELIMINARY

**Problems Definition.** In both the TSP and CVRP, the instance is defined on a complete graph $\mathcal{G} = (\mathcal{V}, \mathcal{E})$, where $\mathcal{V} = \{n_0, n_1, \ldots, n_v\}$ denotes the set of nodes and $\mathcal{E}$ the set of edges. The cost $d(n_i, n_j)$ associated with each edge is typically defined as the Euclidean distance between nodes $n_i$ and $n_j$. In TSP, the objective is to find a minimum-cost Hamiltonian circuit that visits all nodes exactly once. In CVRP, $n_0$ represents the depot, and the remaining nodes correspond to customers with non-negative demands $d_i$. The goal is to construct a set of depot-rooted routes $\tau = \{q_1, \ldots, q_N\}$ such that each customer is visited exactly once, the total demand on each route does not exceed the vehicle capacity $C$, and the overall travel cost is minimized.

**Diffusion Models.** Diffusion models have recently emerged as powerful generative approaches, achieving favorable results in domains such as image synthesis (Sohl-Dickstein et al., 2015; Ho et al., 2020; Rombach et al., 2022; Dhariwal & Nichol, 2021), audio generation (Huang et al., 2023; Kong et al.), and drug design (Schneuing et al., 2024; Xie et al., 2022). Diffusion models operate by corrupting clean inputs with noise and denoise them to reconstruct the original data. Typically, given a clean input $x_0$, the noise process perturbs it by adding Gaussian noise to obtain a noisy sample $x_t$ at time step $t$, following

$$q(x_t \mid x_0) = \mathcal{N}(x_t; \sqrt{\alpha_t}x_0, (1 - \alpha_t)\mathbf{I}), \tag{1}$$

where $\alpha_t \in (0, 1)$ controls the noise level. The model can capture the structure of the target distribution by extracting meaningful signals from noisy states and recovering the original data.

**GFlowNet.** Generative Flow Network (GFlowNet) is a variational inference algorithm that treat sampling from a target probability distribution as a sequential decision-making process (Bengio et al., 2021; 2023), and is widely applied to structured generation tasks, including molecule design (Nica et al., 2022; Jain et al., 2022; Zhu et al., 2023), neural architecture search (Soin et al., 2024; Ikram et al.), and combinatorial optimization (Zhang et al., 2023; a; Chen & Mauch). The objective of GFlowNet is to learn a policy that samples objects in proportion to a given reward distribution. From a high-level perspective, a GFlowNet consists of two complementary components: forward sampling policy and backward policy. The forward pass includes the source flow $Z(\mathcal{G}; \theta)$ and a forward policy $P_F(\tau; \theta)$ that governs trajectory construction from initial to terminal states. The backward pass includes the reward function $R(\tau)$ and a backward policy $P_B(\tau)$ that defines reverse transitions. Together, these bidirectional passes form the basis for training via trajectory balance objectives (Malkin et al., 2022), defined as:

$$\mathcal{L}_{\text{TB}}(\tau; \theta) = \left(\log \frac{Z(\mathcal{G}; \theta)P_F(\tau; \boldsymbol{\theta})}{R(\tau)P_B(\tau)}\right)^2. \tag{2}$$

## 4 DIFFUSION-ENHANCED GFLOWNET

We introduce Diffusion-Enhanced GFlowNet (DEG), a framework designed to overcome a core limitation of existing GFlowNet-based VRP solvers, namely, their inability to effectively expand flow toward high-reward regions. DEG integrates GFlowNet with diffusion by leveraging edge-specific backward signals, which serve as structured noise basis for the diffusion process. A stochastic noise

schedule perturbs these signals, which are subsequently denoised within the GFlowNet paradigm, enabling richer exploration and more sufficient flow expansion. To further strengthen scalability and inference quality across varying problem sizes, DEG incorporates a Graph-Scale Adapter (GSA) decoder, which dynamically aligns spatial statistics across different graph scales.

## 4.1 Edge-specific Backward Signals and Noise Schedule

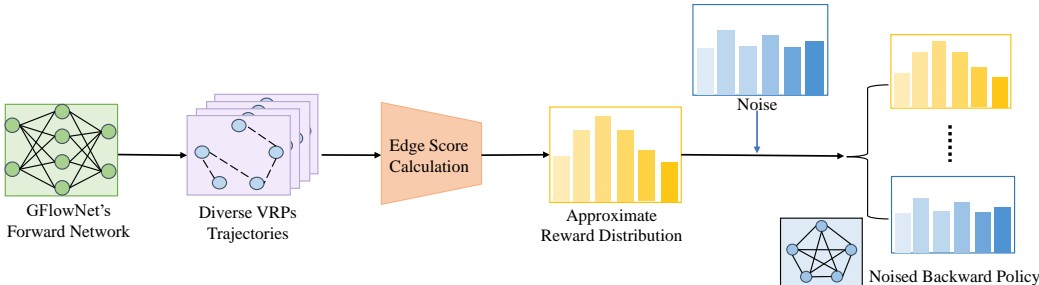

Figure 1: Illustration of Edge-specific Backward Signals Generation and Noise Schedule.

Figure 1 illustrates the generation of edge-specific backward signals and noise schedule. We first use GFlowNet to generate a diverse set of trajectories and assign standardized scores to each edge, estimating an approximate reward distribution that serves as edge-specific backward signals. Although the individual trajectories produced during GFlowNet sampling may not be optimal, their collective information forms a robust reward distribution that guides the diffusion learning process. A noise schedule then injects controlled noise into this distribution to derive a noised backward policy.

By leveraging the inherent diversity of GFlowNet, the model can generate a wide range of trajectories, after which the edge score calculation is applied. The edge score calculation begin by normalizing the reward of each trajectory $\tau_k \in \mathcal{T}$ using the following formulation:

$$-\log R(\tau_k) = D(\tau_k) - \frac{1}{m} \sum_{l=1}^{m} D(\tau_l), \tag{3}$$

where $D(\tau_k)$ is the total distance of trajectory $\tau_k$, defined as: $D(\tau_k) = \sum_{l=1}^{N-1} d(q_l, q_{l+1})$. Here, $d(q_l, q_{l+1})$ denotes the Euclidean distance between nodes $q_l$ and $q_{l+1}$. A higher reward $R(\tau_k)$ corresponds to a shorter, higher-quality trajectory. We define the reward of an edge $e$ as the average reward of the trajectories in which it appears:

$$R(e) = \frac{1}{|\{\tau_k \mid \tau_k \in \mathcal{T}, \ e \in \tau_k\}|} \sum_{\tau_k \in \mathcal{T}, \ e \in \tau_k} R(\tau_k), \tag{4}$$

where the denominator counts the number of trajectories containing edge $e$. This edge-level reward $R(e)$ reflects the global significance of edge $e$ across the sampled solution space. To standardize $R(e)$ across different instances, we use z-score normalization (Montgomery & Runger, 2010): $\hat{R}(e) = \frac{R(e) - \mu_R}{\sigma_R}$, where $\mu_R$ and $\sigma_R$ denote the mean and standard deviation of edge rewards within the instance. To convert the standardized scores into bounded confidence values, we apply the cumulative distribution function (CDF) (Ross, 2014) of the standard normal distribution:

$$x_0 = \Phi\left(\hat{R}(e)\right), \tag{5}$$

where $\Phi(\cdot)$ denotes the CDF of the standard normal distribution. The value $x_0$ represents an approximate reward distribution that provides edge-specific backward signals and serves as the basis for adding noise to integrate diffusion into GFlowNet. The noise schedule then injects Gaussian noise at each training step to obtain a noised backward policy $x_t$:

$$x_t = \sqrt{\alpha_t} \cdot x_0 + \sqrt{1 - \alpha_t} \cdot \epsilon, \quad \epsilon \sim \mathcal{N}(0, 1), \tag{6}$$

where $\alpha_t = \prod_{k=1}^{t}(1 - \beta_k)$, and $\beta_k \in (0, 1)$ is the corruption ratio at diffusion step $k$. To ensure full transformation of the clean signal into noise by the final training iteration $I$, $\beta_k$ should be defined

such that $\prod_{k=1}^{I}(1-\beta_k) \approx 0$. We use continuous Gaussian noise to match the inherently continuous nature of the backward policy in GFlowNet.

**Effect of Edge-specific Backward Signals and Noise Schedule in High Reward Region Exploration.** Our model DEG employs a noise schedule where the corruption level in the backward policy increases progressively throughout training. This utilizes GFlowNet's intrinsic flow balance to navigate the exploitation-exploration trade-off. DEG is trained using the Trajectory Balance (TB) objective (Malkin et al., 2022). Since $Z(\mathcal{G};\theta)$ is fixed for a given graph $\mathcal{G}$, minimizing $\mathcal{L}_{\text{TB}}$ in Eq. 2 encourages the forward policy to satisfy:

$$P_F(\tau) \propto R(\tau) \cdot P_B(\tau). \tag{7}$$

Early in training, when the noise level is low and the solutions lie in low-reward regions, the noised backward policy $P_B(\tau)$ derived from $x_t$ closely approximates the reward distribution. Under this condition, the forward policy simplifies to:

$$P_F(\tau) \propto R(\tau)^2, \tag{8}$$

which reinforces high-reward trajectories, thereby promoting exploitation and accelerating learning. As training progresses, the solutions lie in multiple high-reward regions and the noise level increases, $P_B(\tau)$ becomes increasingly diffuse due to the increasing noise component, approaching a high-entropy distribution. Since the injected Gaussian noise introduces no directional bias, $P_B(\tau)$ can be considered approximately constant at this stage, and the forward policy converges to:

$$P_F(\tau) \propto R(\tau). \tag{9}$$

This facilitates better exploration and promotes stronger flow expansion in high-reward regions, guiding the model to adequately distribute probability across promising solution routes. Thus, our noise schedule approach induces a natural shift in the model's learning target from $R(\tau)^2$ to $R(\tau)$, which corresponds to a gradual transition from exploitation in low-reward regions to exploration in high-reward regions, enabling a unified mechanism for learning a robust and generalizable policy across varying levels of uncertainty. We include experiments in the Appendix A.1 to empirically validate this mechanism.

## 4.2 DIFFUSION-ENHANCE GFLOWNET FRAMEWORK

In our framework, the model learns to construct solutions by drawing on both the noisy backward policy and the reward within the GFlowNet paradigm, which denoises the noised backward policy during training by extracting its informative weights, and using them to enhance forward-policy learning. The loss over a set of trajectories $\mathcal{T} = \{\tau_1, \tau_2, \ldots, \tau_m\}$, based on the denoised forward policy which estimates reward distribution, is defined as:

$$\mathcal{L}(\mathcal{T};\theta) = \sum_{k=1}^{m} \left( \log \frac{Z(\mathcal{G};\theta)P_F(\tau_k;\boldsymbol{\theta})}{R(\tau_k)P_B(\tau_k)} \right)^2. \tag{10}$$

Here, source flow $Z(\mathcal{G};\theta)$ is a learnable partition function derived from Eq. 17, and reward $R(\tau_k)$ is defined in Eq. 3. The forward probability $P_F(\tau_k;\boldsymbol{\theta})$ is computed as:

$$P_F(\tau_k;\boldsymbol{\theta}) = \prod_{t=1}^{N} P_F(s_t|s_{t-1};\boldsymbol{\theta}), \tag{11}$$

where $s_t = \{q_1, q_2, \ldots, q_t\}$ denotes the state at step $t$ during the construction (or destruction) of the trajectory. The term $P_F(s_t|s_{t-1})$ is obtained from the forward policy $\eta(\mathcal{G}^*, \boldsymbol{\theta})$, which is derived from the GFlowNet forward network described in Sec. A.2.

Moreover, $\eta(\mathcal{G}^*, \boldsymbol{\theta})$ can be viewed as a *heatmap* over all edges of the current graph: it is a $|V| \times |V|$ probability matrix where each entry $\eta(i,j)$ gives the likelihood of selecting the edge $(v_i, v_j)$ from the current node $v_i$ to a candidate node $v_j$. During decoding, the next node of the route is sampled according to these edge probabilities. After each selection, the graph state $\mathcal{G}^*$ is updated by marking the chosen node as visited and updating feasibility masks (e.g., vehicle capacity), and the process is repeated until all customers are visited and a complete route is constructed.

Similarly, the backward transition probability $P_B(\tau_k)$ in Eq. 10 is computed as:

$$P_B(\tau_k; \boldsymbol{\theta}) = \prod_{k=1}^{N} P_B(s_{k-1}|s_k; \boldsymbol{\theta}), \quad (12)$$

where $P_B(s_{t-1} \mid s_t; \boldsymbol{\theta})$ is derived from the noised backward policy $x_t$. Specifically, for a partial route state $s_t = \{q_1, q_2, \ldots, q_t\}$, this corresponds to removing the last node $q_t$ from the current partial route, which eliminates the edge $(q_t, q_{t-1})$. Thus $P_B(s_{t-1} \mid s_t; \boldsymbol{\theta})$ is taken directly from the value of $x_t$ at the position $(q_t, q_{t-1})$.

### 4.3 GRAPH-SCALE ADAPTER FOR INFERENCE

During training, we adopt GFlowNet's inherent sampling strategy to generate diverse trajectories to provide edge-specific signal. This enables efficient sampling and strengthens training effectiveness, with a dedicated model trained for specific problem size. However, this strategy does not generalize well at inference time due to scale mismatches between training and test instances.

Specifically, the DEG forward policy learns an approximate reward distribution guided by feedback from the noisy backward flow. However, this design may increase the risk of overfitting to spatial patterns specific to the training scale, such as average neighbor distances and node observation radii defined by the sparsity factor $w$. For example, when trained on 100-node instances, the model implicitly learns to solve instance with an average neighbor distance of approximately $\frac{1}{\sqrt{100}}$ and a node observation distance of $\frac{1}{\sqrt{w}}$. When evaluated on a 500-node instance, however, the average neighbor distance drops to $\frac{1}{\sqrt{500}}$, while the observation distance becomes $\frac{1}{\sqrt{5w}}$. This spatial inconsistency hampers generalization and often leads to suboptimal inference, with the best outcomes of large-scale instances appear at earlier training stages, when the model has not yet overfitted and its estimation of spatial structures remains effective and less biased. We empirically validate this behavior, with details provided in Appendix B.1.

To mitigate this issue, we propose the *Graph-Scale Adapter* (GSA), which aligns the spatial patterns of test instances to those of fixed-size training instances. GSA introduces a scaling factor $\alpha$ that adjusts node distances and sparsity factor in the test graph to match the spatial characteristics of the training graph. The scaling factor is defined as: $\alpha \propto \sqrt{\frac{|\mathcal{V}_{\text{test}}|}{|\mathcal{V}_{\text{train}}|}}$. This adjustment ensures structural consistency across different scales. For instance, if the model is trained on 100-node graphs and evaluated on 500-node graphs, applying GSA ensures that the average neighbor distances and node observation radius remain consistent, approximately $\frac{1}{\sqrt{100}}$ and $\frac{1}{\sqrt{w}}$. To assess the effectiveness of GSA, we apply it during inference on instances of varying sizes. As shown in Appendix B.1, GSA effectively alleviates spatial mismatches caused by scale-specific overfitting and significantly improves the model's ability to generalize across a broad range of instance sizes.

## 5 EXPERIMENTS

We present experimental results to evaluate the performance of DEG. We first assess its effectiveness on both synthetic instances for TSP and CVRP following previous works (Kim et al., 2025; Zhang et al., 2025). Since our model integrates GFlowNet with diffusion, we primarily compare it against GFlowNet-based and diffusion-based baselines. We then evaluate the model's generalization ability on public benchmarks, including TSPLib (Arnold et al., 2019) and CVRPLib (Reinelt, 1991). Finally, we conduct ablation studies to examine the contribution of individual components in DEG. Details of the dataset and hyperparameters are provided in Appendix B.2.

### 5.1 COMPARISON STUDY ON TSP

As prior diffusion-based work has primarily focused on the TSP (Sun & Yang, 2023; Li et al., 2023; 2024; Wang et al., 2025) in the VRP domain, we first evaluate DEG on synthetic TSP instances. Besides the heuristic solver LKH (Helsgaun, 2000), we mainly compare it against both GFlowNet-based solvers including GFACS (Kim et al., 2025) and AGFN (Zhang et al., 2025), and diffusion-based solvers including DIFUSCO (Sun & Yang, 2023), T2T (Li et al., 2023), Fast T2T (Li et al.,

Table 1: Comparison of performance and runtime on TSP with varying node sizes. Obj. denotes the total route length; Gap (%) is computed relative to LKH; and Time (s) indicates the average time required to solve a single instance. For all metrics, lower values indicate better results.

| Method | 500 | | | 1000 | | | 3000 | | |
|--------|------|---------|--------|------|---------|--------|------|---------|--------|
| | Obj. | Time(s) | Gap(%) | Obj. | Time(s) | Gap(%) | Obj. | Time(s) | Gap(%) |
| LKH | 16.55 | 31.80 | – | 23.14 | 97.62 | – | 40.31 | 776.23 | – |
| GFACS (AISTATS'25) | 22.97 | 8.43 | 38.79 | 38.58 | 18.90 | 66.72 | 69.63 | 81.15 | 72.74 |
| AGFN (ICLR'25) | 19.04 | 0.25 | 15.05 | 27.10 | 0.68 | 17.11 | 55.37 | 2.91 | 37.36 |
| DIFUSCO (NeurIPS'23) | 19.12 | 3.52 | 15.17 | 27.70 | 13.60 | 19.71 | 50.39 | 125.02 | 25.01 |
| T2T (NeurIPS'23) | 18.91 | 3.18 | 14.26 | 27.31 | 12.95 | 18.02 | 56.99 | 115.71 | 41.38 |
| Fast T2T (NeurIPS'24) | 18.76 | 0.37 | 13.35 | 27.25 | 1.54 | 17.76 | 57.73 | 4.76 | 43.22 |
| DEITSP (KDD'25) | 18.64 | 0.42 | 12.63 | 27.11 | 1.68 | 17.16 | 54.37 | 5.35 | 34.81 |
| DEG | **18.47** | 0.28 | **11.60** | **26.46** | 0.72 | **14.35** | **47.82** | 3.06 | **18.63** |

| Method | 5000 | | | 8000 | | | 10000 | | |
|--------|------|---------|--------|------|---------|--------|------|---------|--------|
| | Obj. | Time(s) | Gap(%) | Obj. | Time(s) | Gap(%) | Obj. | Time(s) | Gap(%) |
| LKH | 51.84 | 1608.11 | – | 65.89 | 4123.37 | – | 75.78 | 5350.91 | – |
| GFACS (AISTATS'25) | 97.93 | 143.19 | 88.91 | 120.76 | 275.22 | 83.28 | 140.84 | 326.04 | 85.85 |
| AGFN (ICLR'25) | 80.93 | 6.65 | 56.12 | 118.53 | 13.09 | 79.89 | 143.85 | 14.07 | 89.83 |
| DIFUSCO (NeurIPS'23) | – | – | – | – | – | – | – | – | – |
| T2T (NeurIPS'23) | – | – | – | – | – | – | – | – | – |
| Fast T2T (NeurIPS'24) | – | – | – | – | – | – | – | – | – |
| DEITSP (KDD'25) | – | – | – | – | – | – | – | – | – |
| DEG | **63.64** | 6.82 | **22.57** | **83.13** | 14.97 | **26.16** | **94.67** | 16.47 | **24.94** |

*"–" indicates that the model was unable to produce a result for a given instance either within the 90-minute time limit or due to GPU memory constraints.*

2024), and DEITSP (Wang et al., 2025). All models are trained on 100-node instances. To ensure fair comparison, and due to space constraints, Table 1 reports the performance of models *without* 2-opt post-processing. Following (Sun & Yang, 2023; Li et al., 2023; 2024; Wang et al., 2025), results with 2-opt post-processing are provided in Appendix B.3 and Table 7 for completeness. We also provide the comparison on small-scale instances in Appendix B.4 and Table 8.

As shown in Table 1, DEG outperforms all GFlowNet-based and diffusion-based solvers in terms of solution quality. Specially, it achieves improvements of up to 67.43%, 34.19%, 5.10%, 16.10%, 17.17%, and 12.05% over GFACS, AGFN, DIFUSCO, T2T, Fast T2T, and DEITSP, respectively. Notably, the performance gap between our model and other baselines becomes more significant as the instance size increases.

## 5.2 COMPARISON STUDY ON CVRP

We present results on the CVRP instances, focusing primarily on comparisons with GFlowNet-based solvers, as no existing diffusion-based model is capable of solving CVRP. Besides the heuristic solver LKH (Helsgaun, 2000), we compare DEG against GFlowNet-based solvers including GFACS (Kim et al., 2025) and AGFN (Zhang et al., 2025), as well as other neural solvers such as LEHD (Luo et al., 2023) and POMO (Kwon et al., 2020) to ensure comprehensive evaluation. For LKH, we use the setting of 1000-iteration on 3000-, 5000-, 8000- and 10000-node instances, as larger-scale problems require substantial computational time. Also, all neural models are trained on 100-node synthetic instances, and the routes are evaluated directly without post-processing. Experimental results on small-scale instances are further provided in Appendix B.4, and the corresponding comparisons are summarized in Table 8.

As shown in Table 2, DEG achieves better results than GFACS by 13.63%, 18.43%, 7.92%, 8.33%, 12.58% and 21.83% on the 500-, 1000-, 3000-, 5000-, 8000- and 10000-node instances, and outperforms AGFN by 1.82%, 3.21%, 1.86%, 2.00%, 1.05% and 1.35% on the same instance sizes. When compared to Transformer-based AR solvers, NAR methods often struggle to match their performance without the aid of improvement techniques, partly due to inherent limitations in network depth and complexity. Without relying on additional post-processing techniques, our model DEG is able to deliver competitive performance on large-scale instances, while also offering notable advantages in inference efficiency. Compared to state-of-the-art AR mode-LEHD, which adopts a

Table 2: Comparison of performance and runtime on CVRP with varying node sizes. Obj. denotes the total route length; Gap (%) is computed relative to LKH; and Time (s) indicates the average time required to solve a single instance. For all metrics, lower values indicate better results.

| Method | 500 | | | 1000 | | | 3000 | | |
|--------|------|---------|--------|--------|---------|--------|---------|---------|--------|
| | Obj. | Time(s) | Gap(%) | Obj. | Time(s) | Gap(%) | Obj. | Time(s) | Gap(%) |
| LKH | 63.14 | 134.40 | – | 119.52 | 638.40 | – | 380.26* | 671.68 | – |
| GFACS (AISTATS'25) | 80.77 | 10.43 | 27.92 | 158.97 | 22.07 | 33.01 | 393.27 | 130.81 | 3.42 |
| AGFN (ICLR'25) | 71.05 | 0.43 | 12.53 | 133.97 | 0.75 | 12.09 | 368.97 | 3.84 | -2.97 |
| LEHD (NeurIPS'23) | **64.68** | 0.52 | **2.44** | **122.85** | 10.15 | **2.79** | **355.95** | 33.75 | **-6.39** |
| POMO (NeurIPS'20) | 80.21 | 0.50 | 27.04 | 164.17 | 0.92 | 37.36 | 1094.48 | 7.01 | 187.82 |
| POMO(*8) (NeurIPS'20) | 75.45 | 0.67 | 19.50 | 135.74 | 3.13 | 12.74 | 704.99 | 18.26 | 85.40 |
| DEG | 69.76 | 0.45 | 10.48 | 129.67 | 0.80 | 8.49 | 362.11 | 4.08 | -4.77 |

| Method | 5000 | | | 8000 | | | 10000 | | |
|--------|--------|---------|--------|--------|---------|--------|---------|---------|--------|
| | Obj. | Time(s) | Gap(%) | Obj. | Time(s) | Gap(%) | Obj. | Time(s) | Gap(%) |
| LKH(1000) | 642.95* | 2263.34 | – | – | – | – | – | – | – |
| GFACS (AISTATS'25) | 650.62 | 262.63 | 1.19 | 1038.30 | 529.04 | – | 1439.19 | 607.26 | – |
| AGFN (ICLR'25) | 608.58 | 7.42 | -5.35 | 917.24 | 15.24 | – | 1140.46 | 18.58 | – |
| LEHD (NeurIPS'23) | 603.89 | 96.75 | -6.08 | 938.06 | 336.70 | – | 1213.13 | 556.88 | – |
| POMO (NeurIPS'20) | 1438.97 | 19.32 | 123.81 | – | – | – | – | – | – |
| POMO(*8) (NeurIPS'20) | 1381.63 | 41.86 | 114.89 | – | – | – | – | – | – |
| DEG | **596.43** | 10.22 | **-7.24** | **907.65** | 20.32 | **–** | **1125.04** | 24.95 | **–** |

*"–" indicates that the model was unable to produce a result for a given instance either within the 90-minute time limit or due to GPU memory constraints. * denotes LKH run with 1,000 iterations due to time limits.*

Table 3: Comparison of performance on TSPLib. Gap (%) is computed relative to existing best-known solutions, and lower values indicate better results.

| TSP | GFACS | AGFN | DEG | DIFUSCO* | T2T* | Fast T2T* | DEITSP* | DEG* |
|---------|-------|-------|--------|----------|------|-----------|---------|--------|
| 50-200 | 29.60 | 19.41 | **14.37** | 0.96 | 0.37 | 0.26 | 0.81 | **0.21** |
| 200-1000 | 51.67 | 25.51 | **17.68** | 3.45 | 1.84 | 1.26 | 1.46 | **1.03** |

*"–" indicates that the model was unable to produce a result for a given instance either within the 90-minute time limit or due to GPU memory constraints. * represents using two-opt after trajectories are generated.*

heavy decoder transformer network, our model reduces the objective value by 1.24% 3.28% and 7.26% on the 5000-, 8000- and 10000-node instances, respectively, and reduces inference time by 13.46%, 92.61%, 87.91%, 87.37%, 93.96% and 95.58% on 500-, 1000-, 3000-, 8000- and 5000-scales. Furthermore, our model consistently outperforms POMO in objective value and achieves shorter inference time across all instance sizes. Overall, our model achieves competitive results against GFlowNet-based solvers and other neural models, and offers significantly shorter inference time, particularly on large-scale instances. Notably, DEG achieves better performance than both LEHD and POMO on real-world datasets, as detailed in Sec. 5.3 and Table 4.

## 5.3 INFERENCE ON REAL-WORLD DATASET

To assess the generalization ability of our model, we evaluate it on real-world benchmark datasets, TSPLib (Arnold et al., 2019) and CVRPLib (Reinelt, 1991).

Following prior works (Sun & Yang, 2023; Li et al., 2023; 2024; Wang et al., 2025), we evaluate our model on TSPLib with test instances ranging from 50 to 200 nodes and from 200 to 1000 nodes, respectively. We adopt the same experimental setup and baseline methods as described in Section 5.1. The results are reported in Table 3 in the Appendix. Our model outperforms GFACS and AGFN in gap by 51.45% and 25.97% on 200-500 node instances, and 65.78% and 30.69% on 500-1000 node instances, respectively. Compared to the diffusion-based models with two-opt post-processing, DEG achieves superior gap performance over DIFUSCO, DEITSP, T2T, and Fast T2T by 88.59%, 84.56%, 72.73% and 74.07% on 200-500 node instances, and 70.72%, 45.11%, 19.84% and 30.82% on 500-1000 node instances. Overall, our model outperforms both GFlowNet-based models and diffusion-based solvers on TSPLib, across both 50-200 and 200-1000 instances.

We test CVRPLib containing instance with the number of node ranging from 3000 to 16000. The experiment setting and baselines are same with comparison on CVRP benchmarks. As shown in

Table 4: Comparison of performance on CVRPLib. Gap (%) is computed relative to existing best-known solutions, and lower values indicate better results.

| Gap(%) | L1 (3k) | L2 (4k) | A1 (6k) | A2 (7k) | G1 (10k) | G2 (11k) | B1 (15k) | B2 (16k) |
|--------|---------|---------|---------|---------|----------|----------|----------|----------|
| GFACS | 120.21 | 1800.04 | 114.37 | 238.19 | 278.11 | 2242.09 | 348.54 | 2634.74 |
| AGFN | 1144.20 | 27.88 | 29.64 | 24.58 | 142.65 | 27.21 | 20.63 | 160.24 |
| LEHD | **14.04** | 26.30 | 18.90 | 26.40 | 27.23 | 38.45 | 35.94 | 40.76 |
| POMO | 75.30 | 78.16 | 112.27 | 159.22 | – | – | – | – |
| **DEG** | 19.64 | **26.08** | **14.95** | **23.80** | **23.18** | **24.41** | **17.04** | **23.47** |

Table 5: Ablation Study on TSP and CVRP. Obj. denotes the total route length, and Gap (%) is computed relative to LKH; lower values indicate better results.

| TSP | 500 | | 1000 | | 3000 | | 5000 | | 8000 | | 10000 | |
|-----|------|--------|------|--------|------|--------|------|--------|------|--------|------|--------|
| | Obj. | Gap(%) | Obj. | Gap(%) | Obj. | Gap(%) | Obj. | Gap(%) | Obj. | Gap(%) | Obj. | Gap(%) |
| Backbone | 21.96 | 32.69 | 35.41 | 53.03 | 64.82 | 60.80 | 91.66 | 76.81 | – | – | – | – |
| + Diffusion | 18.54 | 12.02 | 26.74 | 15.56 | 48.45 | 20.19 | 64.97 | 25.33 | – | – | – | – |
| + GSA | **18.47** | **11.60** | **26.46** | **14.35** | **47.82** | **18.63** | **63.64** | **22.57** | **83.13** | **26.16** | **94.67** | **24.94** |

| CVRP | 500 | | 1000 | | 3000 | | 5000 | | 8000 | | 10000 | |
|------|------|--------|------|--------|------|--------|------|--------|------|--------|------|--------|
| | Obj. | Gap(%) | Obj. | Gap(%) | Obj. | Gap(%) | Obj. | Gap(%) | Obj. | Gap(%) | Obj. | Gap(%) |
| Backbone | 76.88 | 21.76 | 148.61 | 24.34 | 389.18 | 2.35 | – | – | – | – | – | – |
| + Diffusion | 70.86 | 12.23 | 132.65 | 10.74 | 368.99 | -2.96 | – | – | – | – | – | – |
| + GSA | **69.76** | **10.48** | **129.67** | **8.49** | **362.11** | **-4.77** | **596.43** | – | **907.65** | – | **1125.04** | – |

*"–" indicates that the model was unable to produce a result for a given instance either within the 90-minute time limit or due to GPU memory constraints.*

Table 4, DEG achieves consistently lower optimality gaps than all baseline methods across a wide range of real-world CVRPLib instances. Compared to GFACS, DEG reduces the gap by $83.66\%$, $98.55\%$, $86.93\%$, $90.01\%$, $91.67\%$, $98.91\%$, $95.11\%$, and $99.11\%$ on the L1, L2, A1, A2, G1, G2, B1, and B2 instances, respectively. Compared to AGFN, our model reduces the gap by $39.89\%$, $6.46\%$, $49.56\%$, $3.17\%$, $83.75\%$, $10.29\%$, $17.40\%$, and $85.35\%$ on the same set of instances. Relative to LEHD, DEG achieves gap improvements of $0.84\%$ on L2, $20.89\%$ on A1, $9.85\%$ on A2, $14.87\%$ on G1, $36.51\%$ on G2, $52.59\%$ on B1, and $42.42\%$ on B2, while LEHD performs slightly better on L1. Finally, compared to POMO, our model significantly reduces the gap by $73.92\%$, $66.63\%$, $86.68\%$, and $85.05\%$ on L1, L2, A1, and A2. These results demonstrate the superior scalability and generalization ability of DEG in solving large-scale real-world CVRP instances.

## 5.4 ABLATION STUDY

**Component Evaluation.** We evaluate the contribution of each key component in DEG through an ablation study. We begin by training the backbone model, which involves learning a pure GFlowNet to generate trajectories, serving as the baseline for subsequent comparison. Next, we integrate the diffusion mechanism into the backbone by incorporating reward-level feedback and noise schedule described in Sec.4.1, in order to evaluate the effectiveness of the diffusion component. Finally, the GSA decoder, proposed in Sec. 4.3, is applied on top of the previous model to assess its impact.

The results are presented in Table 5. Incorporating diffusion into the backbone leads to performance improvements of $15.57\%$, $24.48\%$, $25.25\%$, and $29.12\%$ on the 500-, 1000-, 3000-, and 5000-node instances in TSP, and $7.83\%$, $10.74\%$, and $5.19\%$ on the 500-, 1000-, and 3000-node instances in CVRP. Adding the GSA module on top of the diffusion-enhanced model further improves performance by $1.05\%$, $1.30\%$, and $2.05\%$ on TSP, and by $1.55\%$, $2.25\%$, and $1.86\%$ on CVRP across the same instance sizes. In addition to improving performance, GSA also reduces computational resource consumption by shortening the observation distance and decreasing the number of connections per node. As a result, the model is able to efficiently solve 10000-node TSP instances as well as 5000-, 8000-and 10000-node CVRP instances.

**Diversity Test**. As shown in Table 6, we evaluate diversity on both TSP and CVRP by comparing the trajectories generated by (i) a reinforcement-learning baseline, (ii) a vanilla GFlowNet implementation (the backbone of our method), and (iii) our proposed DEG model. The diversity metric (Prins, 2009) is defined as the average number of edges in one trajectory that do not appear in another, a standard measure of route-level diversity in VRP. In this experiment, all settings are kept identical across models, except for the training paradigm. The results show that DEG consistently achieves

Table 6: Diversity comparison on TSP and CVRP, higher values indicate better results.

| TSP | 100 | 500 | 1000 | 3000 | 5000 | CVRP | 100 | 500 | 1000 | 3000 | 5000 |
|---|---|---|---|---|---|---|---|---|---|---|---|
| RL | 17.60 | 96.07 | 217.20 | 819.34 | 1516.64 | RL | 16.57 | 181.36 | 400.71 | 1210.16 | 2372.48 |
| GFlowNet | 20.36 | 126.95 | 294.30 | 1074.53 | 1895.23 | GFlowNet | 19.33 | 212.27 | 443.73 | 1526.56 | 2705.91 |
| DEG | **21.79** | **128.97** | **302.63** | **1145.79** | **2082.36** | DEG | **20.54** | **217.02** | **471.68** | **1648.82** | **2909.12** |

higher diversity than standard GFlowNets, demonstrating that our diffusion-enhanced design effectively enriches the exploration space.

**Learning Efficiency.** We conducted additional experiments on a fixed problem scale and distribution to assess learning efficiency under controlled settings. Specifically, we used 10,000 uniformly sampled TSP and CVRP instances with size $n = 200$ and trained all models with identical configurations, differing only in the training paradigm (RL, standard GFlowNet, PO (Pan et al., 2025), and our DEG). The trained models were then evaluated on TSP/CVRP test instances at $n = 200$, $n = 500$, and $n = 1000$. The results, reported in Appendix Table 9, show that DEG exhibits strong optimization performance and superior learning efficiency compared to these baselines under matched training conditions.

## 5.5 APPLICABILITY TO OTHER COMBINATORIAL OPTIMIZATION PROBLEMS

To further assess the generalization ability of our proposed DEG beyond routing problems, we extended our experiments to additional combinatorial optimization domains. Specifically, we considered: (1) SMTWTP (Single-Machine Total Weighted Tardiness Problem), a classical single-machine scheduling problem where the objective is to minimize the total weighted tardiness of jobs processed on one machine. (2) BPP (Bin Packing Problem), a packing optimization problem that aims to minimize the number of bins required to accommodate items of varying sizes under capacity constraints, thereby improving space utilization. In Appendix Table 10, regarding SMTWTP, the instance sizes (100, 200, and 500) correspond to the number of jobs, with Obj. denoting total weighted tardiness. Regarding BPP, the instance sizes (120, 250, and 500) correspond to the number of items, with Obj. (%) denoting space utilization rate, i.e., the proportion of total bin capacity effectively used by the packed items. As suggested by those results, the proposed DEG model outperforms both DeepACO (Ye et al., 2023) and GFACS Kim et al. (2025) on these tasks, demonstrating its potential as a general neural combinatorial optimization framework beyond routing.

## 6 CONCLUSION

In this paper, we introduce Diffusion-Enhanced GFlowNet (DEG), a generative framework that integrates GFlowNet with diffusion to overcome key limitations of existing GFlowNet-based solvers for vehicle routing problems (VRPs). Specifically, their insufficient flow expansion in high-reward regions. To enable this integration, DEG leverages the intrinsic diversity of GFlowNet to provide edge-specific backward signals as noise basis for diffusion integration. It then employs the diffusion's noise schedule to inject noise into these signals, and denoises them through GFlowNet paradigm, guiding sufficient flow expansion toward high-reward regions to derive better solutions. Furthermore, to enhance scalability and generalization ability, we incorporate a Graph-Scale Adapter (GSA) that enables effective adaptation across diverse instance sizes. Extensive experiments synthetic benchmarks and real-world datasets, including instances with up to 10,000 nodes, show that DEG delivers favorable performance compared to diffusion-, GFlowNet-based and other neural baselines. These results demonstrate that DEG has strong scalability, efficient inference, and robust generalization for solving VRPs. **We will make the code publicly available**. While DEG outperforms the strong AR model like LEHD in inference time and generalization, one of the **limitations** is that DEG's parallel decoding still falls short in small-scale settings, where AR models excel at fine-grained path construction. Bridging this gap remains an important direction for our future work, which involves enhancing the local decision-making capacity of NAR models. In addition, extending DEG to a wider range of VRP variants is another promising venue we plan to explore.

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

## A  MORE METHODOLOGY DETAIL

### A.1  MECHANISM ANALYSIS

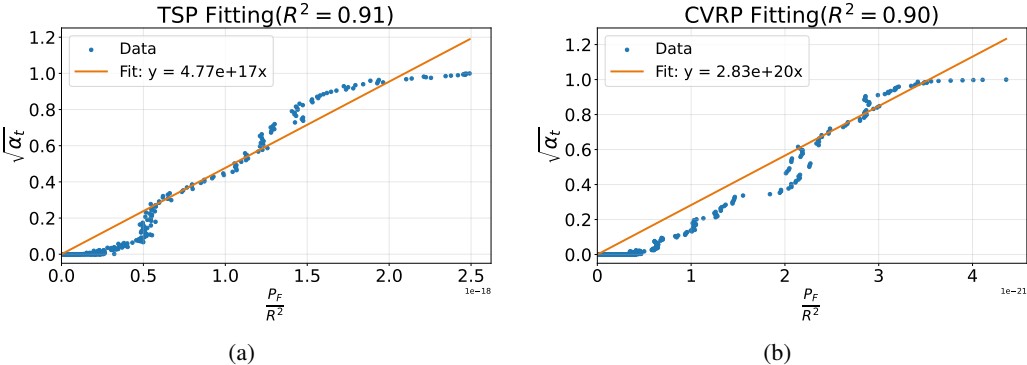

Figure 2: Mechanism Analysis for TSP and CVRP.

We begin with the standard GFlowNet relation from the Eq. 7:

$$P_F(\tau) \propto R(\tau)\,P_B(\tau).$$

In DEG, the backward policy $P_B$ is defined from the diffusion–corrupted reward, as Eq. 6 described:

$$x_t \;=\; \sqrt{\alpha_t}\,R(\tau) + \sqrt{1-\alpha_t}\,\epsilon, \qquad \epsilon \sim \mathcal{N}(0,1).$$

As a result we have $P_B(\tau) \propto x_t$. Substituting this expression into the flow constraint yields

$$P_F(\tau) \;\propto\; R(\tau)\Big(\sqrt{\alpha_t}\,R(\tau) + \sqrt{1-\alpha_t}\,\epsilon\Big). \tag{13}$$

We expand the expression,

$$P_F(\tau) \;\propto\; \sqrt{\alpha_t}\,R(\tau)^2 \;+\; \sqrt{1-\alpha_t}\,R(\tau)\,\epsilon, \qquad \epsilon \sim \mathcal{N}(0,1).$$

Dividing both sides by $R(\tau)^2$ gives

$$\frac{P_F(\tau)}{R(\tau)^2} \;\propto\; \sqrt{\alpha_t} \;+\; \sqrt{1-\alpha_t}\,\frac{\epsilon}{R(\tau)}.$$

Taking expectation over the diffusion noise yields

$$\mathbb{E}_\epsilon\!\left[\frac{P_F(\tau)}{R(\tau)^2}\right] \;\propto\; \sqrt{\alpha_t} \;+\; \sqrt{1-\alpha_t}\,\frac{\mathbb{E}[\epsilon]}{R(\tau)}.$$

Since $\epsilon \sim \mathcal{N}(0,1)$ implies $\mathbb{E}[\epsilon] = 0$, the second term vanishes, and we obtain

$$\mathbb{E}_\epsilon\!\left[\frac{P_F(\tau)}{R(\tau)^2}\right] \;\propto\; \mathbb{E}_\epsilon[\sqrt{\alpha_t}]. \tag{14}$$

We empirically validate this equation directly during training. At each training step $t$, we compute the quantity $\frac{P_F(\tau_t)}{R(\tau_t)^2}$, where $\tau_t$ is the trajectory sampled at step $t$. Every 10 steps, we record the empirical average of this quantity and pair it with the corresponding average diffusion level $\mathbb{E}_\epsilon\big[\sqrt{\alpha_t}\big] \approx \frac{1}{10}\sum_{s=t-9}^{t}\sqrt{\alpha_s}$. This produces a sequence of averaged values

$$\mathbb{E}_\epsilon\!\left[\frac{P_F(\tau)}{R(\tau)^2}\right] \;\approx\; \frac{1}{10}\sum_{s=t-9}^{t}\frac{P_F(\tau_s)}{R(\tau_s)^2},$$

which reflects the behavior of $P_F(\tau)/R(\tau)^2$ throughout training.

As shown in Fig. 2, where $R^2$ denotes the coefficient of determination, values closer to 1 indicate a stronger goodness of fit. In practice, an $R^2$ above 0.9 is generally considered a strong indication of a good linear fit. The coefficient of determination show that the empirical sequence $\mathbb{E}_\epsilon\left[\frac{P_F(\tau)}{R(\tau)^2}\right]$ exhibits a nearly perfect linear relationship with $\mathbb{E}_\epsilon\left[\sqrt{\alpha_t}\right]$, confirming the theoretical prediction:

$$\mathbb{E}\left[\frac{P_F(\tau)}{R(\tau)^2}\right] \propto \sqrt{\alpha_t}.$$

Thus, the assumptions stated in Eq. 7 and Eq. 13 are empirically validated. Consequently, the theoretical interpretation of DEG based on both equations is supported: at the beginning of training, the forward policy behaves according to Eq. 8, emphasizing exploitation in low-reward regions, as illustrated by the fitting in the upper-right area of the two figures (Fig. 2). As training progresses, the policy gradually shifts toward the behavior described in Eq. 9, enabling increased exploration in high-reward regions, which is reflected by the fitting toward the lower-left area in the figures (Fig. 2).

## A.2 FORWARD NETWORK AND TRAINING PARADIGM

Following prior work (Kim et al., 2025; Zhang et al., 2025; Sun & Yang, 2023; Xin et al., 2021), we adopt a Graph Neural Network (GNN) as the forward network $\theta$, incorporating distance information as edge features. GNN employs a sparsification strategy on the input graph $\mathcal{G}$ to reduce computational complexity. Specifically, the graph retains only the $w$ shortest edges per node, resulting in a sparse graph $\mathcal{G}^* = (\mathcal{V}, \mathcal{E}^*)$. The node coordinates $\mathbf{u}_v \in \mathbb{R}^2$ and edge distances $\mathbf{u}_e \in \mathbb{R}$ are first embedded into initial node and edge features $\mathbf{h}_i^0, \mathbf{e}_{ij}^0 \in \mathbb{R}^d$ by learnable linear layers, where $d$ represents the number of hidden units per layers. Then, $\mathbf{h}_i^0$ and $\mathbf{e}_{ij}^0$ are processed by GNN to capture the underlying relationships between nodes and edges. At each layer $l$, node and edge features are updated as:

$$\mathbf{h}_i^{l+1} = \mathbf{h}_i^l + \mathrm{ACT}(\mathrm{BN}(\mathbf{U}^l\mathbf{h}_i^l + \mathcal{A}_{j\in\mathcal{N}_i}(\sigma(\mathbf{e}_{ij}^l) \odot \mathbf{V}^l\mathbf{h}_j^l))), \tag{15}$$

$$\mathbf{e}_{ij}^{l+1} = \mathbf{e}_{ij}^l + \mathrm{ACT}(\mathrm{BN}(\mathbf{P}^l\mathbf{e}_{ij}^l + \mathbf{Q}^l\mathbf{h}_i^l + \mathbf{R}^l\mathbf{h}_j^l)). \tag{16}$$

Here, $\mathbf{U}^l, \mathbf{V}^l, \mathbf{P}^l, \mathbf{Q}^l, \mathbf{R}^l \in \mathbb{R}^{d\times d}$ are learnable parameters of the GNN layers; SiLU (Elfwing et al., 2018) is used as the activation function for all intermediate layers; BN denotes batch normalization; $\sigma$ is the sigmoid function (Han & Moraga, 1995); $\odot$ is the Hadamard product; and $\mathcal{A}$ denotes mean pooling over neighbors.

The denoised forward policy $\eta(\mathcal{G}^*, \boldsymbol{\theta})$ is computed using a multi-layer perceptron (MLP) applied to the final-layer GNN embeddings, followed by a sigmoid activation to normalize outputs. The source flow $Z(\mathcal{G}; \theta)$ is computed from the final-layer edge embedding $e^d$ as follows:

$$Z(\mathcal{G}; \theta) = M \cdot \mathrm{ReLU}(J \cdot e^d + b) + f, \tag{17}$$

where $M$, $J$, $b$, and $f$ are learnable parameters, and ReLU is the activation function.

## B MORE EXPERIMENT DETAIL

### B.1 GSA COMPARISON RESULTS

We validate the behavior that the best outcomes of larger-scale instances often appear at earlier training stages. We evaluate the model on datasets of varying sizes (200, 500, and 1000 nodes), each containing 128 test cases. The model is trained on 100-node instances and evaluated using checkpoints saved at different training epochs, with each epoch comprising 20 training steps. The results are presented as in Fig. 3. As the instance size increases from 200 to 500 and then to 1000 nodes, the performance peak occurs progressively earlier during training. Moreover, the model's performance degrades significantly after reaching its peak, especially on larger-scale instances.

After using GSA, as illustrated in Fig. 4, GSA successfully constrains the optimal inference checkpoints across different instance sizes into a consistent interval. Moreover, as training progresses, the

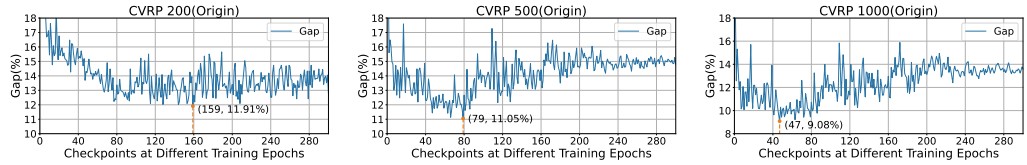

Figure 3: The original performance of DEG test on different scale instance. Gap (%) is computed relative to LKH, and lower values indicate better results.

model shows a steadily improving performance at later checkpoints—unlike the original decoder, which tends to degrade after early peaks. In addition, GSA achieves overall better performance compared to the baseline setup without GSA. These results indicate that GSA mitigates spatial pattern overfitting to the training instance size and enables the model to derive consistent results across varying instance scales.

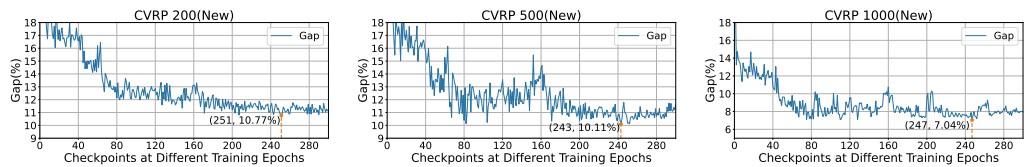

Figure 4: The performance of GSA-based DEG test on different scale instance. Gap (%) is computed relative to LKH, and lower values indicate better results.

## B.2 DATASET AND HYPERPARAMETERS

Following prior work on VRP (Sun & Yang, 2023; Kim et al., 2025; Zhang et al., 2025; b; Wang et al., 2025), we use synthetic datasets for training both TSP and CVRP models. Each TSP instance consists of nodes represented only by their 2D coordinates, sampled independently from the unit square $[0, 1]^2$, without any depot or demand. The objective is to compute the shortest tour that visits each node exactly once and returns to the start. For CVRP, each instance includes a single depot and multiple customer nodes served by a vehicle with a fixed capacity $C = 50$. Node locations (both depot and customers) are uniformly sampled from $[0, 1]^2$. Each customer is assigned a demand independently drawn from a uniform distribution $U[a, b]$, where $a = 1$ and $b = 9$. For training, DEG is trained on 100-node instances for both TSP and CVRP. For evaluation, we generate synthetic test sets containing 128 instances per problem size, ranging from 500 to 10,000 nodes (500, 1,000, 3,000, 5,000, 8,000 and 10,000).

Following standard practice (Sun & Yang, 2023; Wang et al., 2025; Graikos et al., 2022), we adopt a linear noise schedule with $\beta_1 = 10^{-4}$ and $\beta_I = 0.02$. The model is trained for $I = 6000$ steps, using a batch size of 10 instances per step. We use the AdamW optimizer (Loshchilov & Hutter) with an initial learning rate of $10^{-4}$, decayed to $10^{-5}$ via a CosineAnnealingLR schedule (Loshchilov & Hutter, 2022). All experiments are conducted on a system with an NVIDIA H100 NVL GPU and an AMD EPYC 9554 CPU.

## B.3 TSP COMPARISON ON DIFFUSION-BASED SOLVER WITH TWO-OPT

We compare DEG with existing diffusion-based models, all enhanced with Two-Opt post-processing, and the results are summarized in Table 7. Our method, DEG, achieves a objective value reduction of 2.42%, 2.27%, and 2.74% on 500-, 1000-, 3000-node instance over DIFUSCO. Compared to Fast-T2T, DEG yields improvements of 0.88%, 1.66%, and 4.17% on the same instance sizes. Furthermore, DEG outperforms DEITSP by 0.80%, 2.04% on 500-, 1000-, 3000-node instances. Notably, DEG is capable of scaling to larger problem sizes, specifically 5,000 8,000 and 10,000 nodes, whereas other diffusion-based models fail to run due to memory limitations.

Table 7: Comparison of performance on TSP with varying node sizes with two-opt post processing. Obj. denotes the total route length and Gap (%) is computed relative to LKH. For all metrics, lower values indicate better results.

| TSP | 500 | | 1000 | | 3000 | | 5000 | | 8000 | | 10000 | |
|---|---|---|---|---|---|---|---|---|---|---|---|---|
| | Obj. | Gap(%) | Obj. | Gap(%) | Obj. | Gap(%) | Obj. | Gap(%) | Obj. | Gap(%) | Obj. | Gap(%) |
| DIFUSCO (NeurIPS'23) | 17.32 | 4.65 | 24.22 | 4.66 | 43.41 | 7.69 | – | – | – | – | – | – |
| T2T (NeurIPS'23) | 17.14 | 3.56 | 24.18 | 4.49 | 42.97 | 6.60 | – | – | – | – | – | – |
| Fast T2T (NeurIPS'24) | 17.05 | 3.02 | 24.07 | 4.02 | 44.06 | 9.30 | – | – | – | – | – | – |
| DEITSP (KDD'25) | 16.98 | 2.60 | 23.86 | 3.15 | 43.10 | 6.92 | – | – | – | – | – | – |
| DEG | **16.90** | **2.11** | **23.67** | **2.30** | **42.22** | **4.74** | **54.61** | **5.34** | **69.46** | **5.42** | **80.05** | **5.69** |

*"–" indicates that the model was unable to produce a result for a given instance either within the 90-minute time limit or due to GPU memory constraints.*

Table 8: Comparison on small-scale TSP and CVRP instances (100 and 200 Nodes). Obj. denotes the total route length; Gap (%) is computed relative to LKH; and Time (s) indicates the average time required to solve a single instance. For all metrics, lower values indicate better results.

| TSP | 100 | | | 200 | | |
|---|---|---|---|---|---|---|
| | Obj. | Time(s) | Gap(%) | Obj. | Time(s) | Gap(%) |
| LKH | 7.76 | 1.50 | - | 10.62 | 9.80 | - |
| GFACS | 8.80 | 0.53 | 13.54 | 13.20 | 1.15 | 24.29 |
| AGFN | 8.49 | 0.05 | 9.55 | 11.82 | 0.08 | 11.30 |
| DEG | **8.40** | 0.06 | **8.25** | **11.65** | 0.10 | **9.42** |
| DIFUSCO (w/ LS) | 7.88 | 0.42 | 1.68 | 10.80 | 1.53 | 1.70 |
| T2T (w/ LS) | 7.80 | 0.37 | 0.65 | 10.80 | 1.15 | 1.72 |
| DEITSP (w/ LS) | **7.78** | 0.08 | **0.39** | 10.76 | 1.20 | 1.32 |
| Fast T2T (w/ LS) | 7.82 | 0.11 | 0.90 | 10.78 | 1.19 | 1.51 |
| DEG (w/ LS) | 7.79 | 0.07 | 0.52 | **10.76** | 1.15 | **1.32** |

| CVRP | 100 | | | 200 | | |
|---|---|---|---|---|---|---|
| | Obj. | Time(s) | Gap(%) | Obj. | Time(s) | Gap(%) |
| LKH | 15.57 | 18.80 | - | 28.04 | 50.83 | - |
| GFACS | 19.26 | 1.90 | 23.70 | 35.36 | 3.14 | 26.11 |
| AGFN | 17.78 | 0.06 | 14.19 | 31.60 | 0.16 | 12.70 |
| DEG | **17.07** | 0.07 | **9.63** | **31.00** | 0.18 | **10.56** |

## B.4 TSP AND CVRP COMPARISON IN SMALL-SCALE PROBLEMS

To evaluate the performance of DEG on small-scale problems, we additionally conduct experiments on TSP and CVRP instances of 100 and 200 nodes. The comparisons include both diffusion-based and GFlowNet-based baselines; for CVRP, only GFlowNet-based baselines are reported because diffusion-based solvers do not directly apply to this setting. The results, summarized in Table 8, show that DEG consistently surpasses all diffusion-based and GFlowNet-based methods on TSP200, CVRP100, and CVRP200, and achieves competitive or superior performance on TSP100. These findings demonstrate that DEG also maintains strong optimization capability on problems of relatively small scales.

## B.5 MULTIPLE SCALE TRAINING

We conducted additional experiments where DEG is trained on multiple node sizes. Specifically, for both TSP and CVRP, we trained models on instances of size 100, 200, 500, and 1000, and evaluated them on larger test sets with 500, 1000, 3000, 5000, 8000, and 10000 nodes. For each training scale, we report results with and without GSA rescaling.

The results (provided in Table 11 and Table 12) show that DEG maintains strong performance even without GSA, while GSA further strengthens generalization on very large instances. These findings confirm that the scalability of DEG is not dependent on training exclusively at the 100-node scale, and that the model possesses substantial inherent scale robustness.

## B.6 GSA APPLICABILITY.

We applied GSA to two strong frameworks, GFACS and AGFN, to examine whether similar scale-adaptation benefits can be observed. As shown in Table 13 and Table 14, incorporating GSA consistently improves the generalization performance of both models when evaluated on scales different from their training sizes. These results confirm that GSA is a reusable and beneficial component for neural solvers in tackling vehicle routing problems.

Table 9: Comparison of training efficiency on TSP and CVRP. Obj. denotes the total route length, and Gap (%) is computed relative to LKH. For both metrics, lower values indicate better results.

| TSP | 200 | | 500 | | 1000 | |
|---|---|---|---|---|---|---|
| | Obj. | Gap(%) | Obj. | Gap(%) | Obj. | Gap(%) |
| RL | 12.12 | 14.12 | 19.28 | 16.50 | 27.82 | 20.22 |
| GFlowNet | 12.08 | 13.75 | 19.02 | 14.92 | 27.29 | 17.93 |
| PO | 11.97 | 12.71 | 18.93 | 14.38 | 27.11 | 17.16 |
| DEG | **11.91** | **12.15** | **18.79** | **13.53** | **26.80** | **15.82** |

| CVRP | 200 | | 500 | | 1000 | |
|---|---|---|---|---|---|---|
| | Obj. | Gap(%) | Obj. | Gap(%) | Obj. | Gap(%) |
| RL | 32.94 | 17.48 | 72.50 | 14.82 | 133.41 | 11.62 |
| GFlowNet | 32.03 | 14.23 | 71.49 | 13.22 | 132.47 | 10.84 |
| PO | 32.13 | 14.59 | 71.77 | 13.67 | 133.39 | 11.60 |
| DEG | **31.90** | **13.77** | **70.15** | **11.10** | **131.23** | **9.80** |

Table 10: Results on SMTWTP and BPP benchmarks. For SMTWTP, Obj. denotes total weighted tardiness (lower is better); for BPP, Obj. denotes space utilization rate (higher is better). Gap (%) is computed relative to LKH, where lower values indicate better performance.

| SMTWTP | 100 | | 200 | | 500 | |
|---|---|---|---|---|---|---|
| | Obj.↓ | Time(s) | Obj.↓ | Time(s) | Obj.↓ | Time(s) |
| ACO | 361.22 | 0.82 | 1608.26 | 1.70 | 11015.39 | 2.84 |
| DeepACO | 223.36 | 0.93 | 905.19 | 1.81 | 1567.20 | 3.56 |
| GFACS | 40.67 | 1.04 | 119.32 | 2.25 | 559.31 | 3.68 |
| DEG | **37.70** | **0.14** | **103.13** | **0.34** | **510.67** | **0.98** |

| BPP | 120 | | 250 | | 500 | |
|---|---|---|---|---|---|---|
| | Obj.↑ | Time(s) | Obj.↑ | Time(s) | Obj.↑ | Time(s) |
| ACO | 90.65 | 2.95 | 90.36 | 5.85 | 90.17 | 10.00 |
| DeepACO | 93.46 | 2.37 | 93.10 | 5.50 | 93.08 | 10.75 |
| GFACS | 94.68 | 2.38 | 94.03 | 5.56 | 93.65 | 10.77 |
| DEG | **96.94** | **0.27** | **96.30** | **0.58** | **95.37** | **1.38** |

Table 11: Training on different scales for TSP and testing without and with GSA. Obj. denotes the total route length, and Gap (%) is computed relative to LKH. For both metrics, lower is better.

| | 500 | | 1000 | | 3000 | | 5000 | | 8000 | | 10000 | |
|---|---|---|---|---|---|---|---|---|---|---|---|---|
| | Obj. | Gap(%) | Obj. | Gap(%) | Obj. | Gap(%) | Obj. | Gap(%) | Obj. | Gap(%) | Obj. | Gap(%) |
| 100(w/o GSA) | 18.54 | 12.02 | 26.74 | 15.56 | 48.45 | 20.19 | 64.97 | 25.33 | – | – | – | – |
| 200(w/o GSA) | 18.44 | 11.42 | 26.50 | 14.52 | 47.59 | 18.06 | 64.16 | 23.77 | – | – | – | – |
| 500(w/o GSA) | **18.05** | **9.06** | 26.42 | 14.17 | 47.19 | 17.07 | **61.55** | **18.73** | – | – | – | – |
| 1000(w/o GSA) | 18.41 | 11.24 | **25.87** | **11.80** | **46.32** | **14.91** | 63.14 | 21.80 | – | – | – | – |
| 100(w/ GSA) | 18.47 | 11.60 | 26.46 | 14.35 | 47.82 | 18.63 | 63.64 | 22.57 | 83.13 | 26.16 | 94.67 | 24.94 |
| 200(w/ GSA) | 18.33 | 10.76 | 26.34 | 13.83 | 46.91 | 16.37 | 62.16 | 19.91 | 82.19 | 24.74 | 91.09 | 20.20 |
| 500(w/ GSA) | **18.08** | **9.24** | 25.97 | 12.23 | 46.05 | **14.24** | **61.11** | **17.88** | **78.16** | **18.62** | 87.44 | **15.39** |
| 1000(w/ GSA) | 18.28 | 10.45 | **25.87** | **11.80** | **45.09** | 14.86 | 62.31 | 20.20 | 79.06 | 19.99 | 89.58 | 18.21 |

*"–" indicates that the model was unable to produce a result for a given instance either within the 90-minute time limit or due to GPU memory constraints.*

Table 12: Training on different scales for CVRP and testing without and with GSA. Obj. denotes the total route length, and Gap (%) is computed relative to LKH. For both metrics, lower values indicate better results.

| | 500 | | 1000 | | 3000 | | 5000 | | 8000 | | 10000 | |
|---|---|---|---|---|---|---|---|---|---|---|---|---|
| | Obj. | Gap(%) | Obj. | Gap(%) | Obj. | Gap(%) | Obj. | Gap(%) | Obj. | Gap(%) | Obj. | Gap(%) |
| 100 (w/o GSA) | 69.76 | 10.48 | 129.67 | 8.49 | 362.11 | -4.77 | 596.43 | -7.24 | 907.65 | – | 1125.04 | – |
| 200 (w/o GSA) | 67.75 | 7.30 | 127.70 | 6.84 | 362.43 | -4.69 | 580.73 | -9.68 | 987.01 | – | 1094.82 | – |
| 500 (w/o GSA) | **66.76** | **5.73** | 124.70 | 4.34 | 360.80 | -5.12 | 577.24 | -10.22 | 894.98 | – | 1072.27 | – |
| 1000(w/o GSA) | 69.68 | 10.36 | **123.13** | **3.02** | 360.42 | **-5.21** | 568.73 | **-11.54** | 885.18 | – | 1060.53 | – |
| 100 (w/ GSA) | 70.86 | 12.23 | 132.65 | 10.74 | 368.99 | -2.96 | – | – | – | – | – | – |
| 200 (w/ GSA) | 68.36 | 8.37 | 129.50 | 8.35 | 368.85 | -3.00 | – | – | – | – | – | – |
| 500 (w/ GSA) | **66.76** | **5.73** | 125.65 | 5.13 | 367.13 | -3.45 | – | – | – | – | – | – |
| 1000(w/ GSA) | 70.54 | 11.72 | **123.13** | **3.02** | 366.19 | **-3.70** | – | – | – | – | – | – |

*"–" indicates that the model was unable to produce a result for a given instance either within the 90-minute time limit or due to GPU memory constraints.*

Table 13: GSA application to GFACS and AGFN on TSP. Obj. denotes the total route length, and Gap (%) is computed relative to LKH. For both metrics, lower values indicate better results.

| | 500 | | 1000 | | 3000 | | 5000 | | 8000 | | 10000 | |
|---|---|---|---|---|---|---|---|---|---|---|---|---|
| | Obj. | Gap(%) | Obj. | Gap(%) | Obj. | Gap(%) | Obj. | Gap(%) | Obj. | Gap(%) | Obj. | Gap(%) |
| GFACS | 22.97 | 38.79 | 38.58 | 66.72 | 69.63 | 72.74 | 97.93 | 88.91 | 120.76 | 83.28 | 140.84 | 85.85 |
| GFACS w/ GSA | **22.10** | **33.53** | **35.90** | **55.14** | **62.70** | **55.54** | **88.46** | **70.64** | **113.88** | **72.07** | **129.39** | **70.74** |
| AGFN | 19.04 | 15.05 | 27.10 | 17.11 | 55.37 | 37.36 | 80.93 | 56.12 | 118.53 | 79.89 | 143.85 | 89.83 |
| AGFN w/ GSA | **18.87** | **14.02** | **26.92** | **16.34** | **53.26** | **32.13** | **75.95** | **46.51** | **110.52** | **67.73** | **127.76** | **68.59** |

## C   Detailed Introduction of GFlowNet and DEG

### C.1   What is GFlowNet.

The Generative Flow Network (GFlowNet) is probabilistic generative model that treats the process of sampling from a target distribution as a sequential decision-making problem (Bengio et al., 2021; 2023). Formally, a GFlowNet operates on a fully-observed, deterministic Markov Decision Process (MDP) with a set of states $\mathcal{S}$, a set of actions $\mathcal{A} \subseteq \mathcal{S} \times \mathcal{S}$, and a designated initial state $s_0$. Certain states are terminal and have no outgoing actions; we denote the set of terminal states by $\mathcal{X}$. Each terminal state $x \in \mathcal{X}$ is reachable from $s_0$ through at least one (not necessarily unique) sequence of actions.

A complete trajectory is defined as $\tau = (s_0 \rightarrow s_1 \rightarrow \cdots \rightarrow s_n)$, where $s_n \in \mathcal{X}$ and every transition satisfies $(s_i, s_{i+1}) \in \mathcal{A}$. A *forward policy* $P_F(s' \mid s)$ defines the probability distribution over next states from $s$. It induces a distribution over trajectories:

$$P_F(\tau) = P_F(s_0 \rightarrow s_1 \rightarrow \cdots \rightarrow s_n) = \prod_{i=0}^{n-1} P_F(s_{i+1} \mid s_i).$$

Each terminal state $x \in \mathcal{X}$ is associated with a reward $R(x) > 0$, interpreted as an unnormalized probability mass. The learning objective of a GFlowNet is to produce a policy $P_F$ whose induced marginal distribution over terminal states satisfies

$$P_F(x) \propto R(x).$$

In other words, a GFlowNet learns to sample terminal objects with probability proportional to their reward.

**Trajectory Balance (TB).**   The trajectory balance objective (Malkin et al., 2022) introduces a scalar parameter $Z_\theta$ (an estimate of the partition function) and a backward policy $P_B$. For a complete trajectory $\tau = (s_0 \rightarrow \cdots \rightarrow s_n = x)$, the TB loss is

$$\ell_{\mathrm{TB}}(\tau; \theta) = \left( \log \frac{Z_\theta \prod_{i=0}^{n-1} P_F(s_{i+1} \mid s_i; \theta)}{R(x) \prod_{i=0}^{n-1} P_B(s_i \mid s_{i+1}; \theta)} \right)^2 .$$

If the TB loss is driven to zero for all trajectories, the forward policy satisfies the desired reward-proportional condition. In practice, GFlowNets predict the log-probabilities of both forward and backward transitions, as well as the log-flow $F(s)$ and the log-partition constant $\log Z$.

**Training and Exploration.**   TB losses depend on transitions or full trajectories, but do not prescribe how trajectories should be chosen. A common approach is on-policy training, where trajectories are sampled based on the current forward policy $P_F$ (which assigns probabilities to edges) and backward policy $P_B$(which determines either exploration or exploitation during training), and the TB objective is minimized through gradient descent.

**Conditional GFlowNets.**   In many applications, including combinatorial optimization, the reward and admissible actions depend on an input instance. Conditional GFlowNets extend the formulation so that $P_F$ and $P_B$ are conditioned on input features, enabling generalization to unseen instances and supporting tasks where solutions correspond to variable-sized or structured objects.

### C.2   How DEG is Trained and How Solutions Are Constructed.

To make the training and inference process of DEG more accessible to readers who may not be familiar with GFlowNets, we provide a VRP-oriented and intuitive explanation below.

**A more accessible explanation of GFlowNets.**   A Generative Flow Network (GFlowNet) is a model that learns to generate structured objects by viewing the construction process as a sequence of decisions. Its core idea is to enforce a balance of "flow'" across all possible paths, so that the probability of producing each final object is proportional to its reward. In this sense, GFlowNet

Table 14: GSA application to GFACS and AGFN on CVRP. Obj. denotes the total route length, and Gap (%) is computed relative to LKH. For both metrics, lower values indicate better results.

| | 500 | | 1000 | | 3000 | | 5000 | | 8000 | | 10000 | |
| --- | --- | --- | --- | --- | --- | --- | --- | --- | --- | --- | --- | --- |
| | Obj. | Gap(%) | Obj. | Gap(%) | Obj. | Gap(%) | Obj. | Gap(%) | Obj. | Gap(%) | Obj. | Gap(%) |
| GFACS | 80.77 | 27.92 | 158.97 | 33.01 | 393.27 | 3.42 | 650.62 | 1.19 | 1038.30 | – | 1439.19 | – |
| GFACS w/ GSA | **80.26** | **27.11** | **146.76** | **22.79** | **383.50** | **0.85** | **622.67** | **-3.15** | **922.24** | – | **1323.82** | – |
| AGFN | 71.05 | 12.53 | 133.97 | 12.09 | 368.97 | -2.97 | 608.58 | -5.35 | 917.24 | – | 1140.46 | – |
| AGFN w/ GSA | **70.74** | **12.04** | **131.60** | **10.11** | **366.72** | **-3.56** | **602.76** | **-6.25** | **909.15** | – | **1131.58** | – |

is better understood as a training paradigm that specifies how to learn a policy. Due to the flow-balance objective, this paradigm naturally leads to diverse solutions instead of collapsing to a single one, which makes GFlowNets particularly suitable for complex combinatorial problems.

**DEG learns an edge-level heatmap policy.** DEG follows a non-autoregressive GFlowNet paradigm in which the neural network produces, in a single forward pass, an edge-level heatmap over the entire graph. Given the VRP instance, such as node coordinates, customer demands, and depot information, the model outputs a score or unnormalized probability for every directed edge. These edge scores (the heatmap) encode the model's belief about how promising each possible connection is in the final solution. During inference, they guide the construction of the full VRP solution by selecting edges according to the learned heatmap.

**DEG adjusts the policy so that high-reward routes are sampled more often.** Using the GFlowNet training paradigm, the model is optimized so that the probability of sampling a route is proportional to its corresponding reward. This is fundamentally different from reinforcement learning, which aims at locating a single best route; DEG instead learns a distribution over many high-quality routes.

**Diffusion noise guides exploration toward high-reward regions.** DEG modifies the backward-propagated reward signal by injecting diffusion-style noise to modulate the trade-off between exploration and exploitation. Early in training, the noise level is low, enabling exploitation of promising routing structures. As training progresses, the noise level gradually increases, encouraging broader exploration in high-reward regions.

**DEG Training.** Based on above, now we can discuss about how DEG is training, where the training paradigm is:

$$\mathcal{L}(\mathcal{T}; \theta) = \sum_{k=1}^{m} \left( \log \frac{Z(\mathcal{G}; \theta) P_F(\tau_k; \theta)}{R(\tau_k) P_B(\tau_k)} \right)^2 .$$

In this equation, $Z(\mathcal{G}; \theta)$ denotes the source flow, computed from the instance $\mathcal{G}$ through the neural network $\theta$. The forward policy $P_F(\tau_k; \theta)$ is the product of the heatmap entries corresponding to all edges along trajectory $\tau$. The reward $R(\tau_k)$ is defined in Eq. 3. Finally, $P_B(\tau_k)$ is the backward policy introduced in our work, where we inject diffusion-style noise to dynamically control the exploitation–exploration behavior during training.

**Solution Construction.** During training or inference, DEG produces a complete VRP solution directly from the learned edge-level heatmap, without relying on step-by-step autoregressive decisions. Given a VRP instance, the trained model generates, in a single forward pass, an N×N heatmap that assigns a probability or score to every directed edge. This heatmap reflects the model's learned preference for which edges are likely to appear in high-reward solutions.

To construct the set of routes, the heatmap is first masked using VRP feasibility constraints, such as vehicle capacity and visit requirements, to ensure that only valid edges remain selectable. A route is then generated by selecting edges according to their heatmap scores (typically by sampling with probabilities proportional to their heatmap values, or by applying a greedy selection rule to choose the valid edge with highest probability). When the selected edges form a cycle or the vehicle capacity is exhausted, the current route is terminated. The same process is then applied to the remaining unvisited customers until every customer has been assigned to exactly one route.

## C.3  STATISTICAL SIGNIFICANCE OF EXPERIMENT

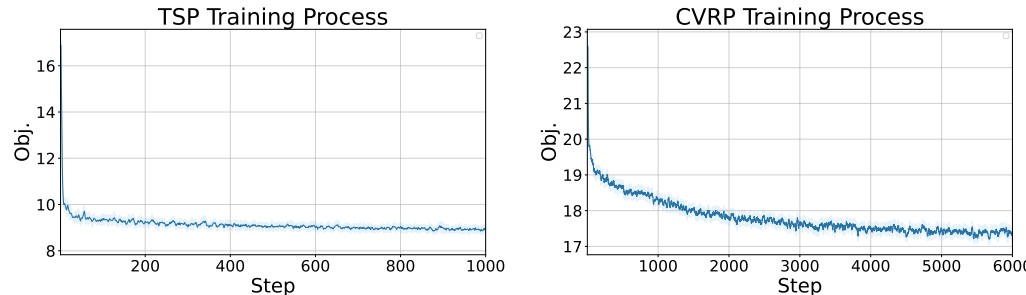

Figure 5: Training Process on TSP and CVRP. The solid line represents the average over five random seeds, while the shaded area indicates the standard deviation. Obj. denotes the total route length, and lower values indicate better results.

To assess the reliability of our training results, we report the mean and standard deviation across five runs with different random seeds. As shown in Fig. 5, the solid lines indicate the average objective values over training steps, while the shaded regions represent the corresponding standard deviations. The consistently narrow shaded areas throughout training suggest low variance across runs, indicating that the training process is stable and the performance improvements are statistically reliable. These results demonstrate the robustness of our method under different initialization conditions and validate the consistency of the observed performance trends.

## D  AUTHOR STATEMENT ON THE USE OF LARGE LANGUAGE MODELS

Large language models (LLMs) were used for language polishing in this paper. Their role was limited to improving grammar, clarity, and readability of the text. All research ideas, methods, analyses, and conclusions are entirely the work of the authors, and the LLMs contributed no conceptual or experimental input.

