# OpenReview forum: "Diffusion-Enhanced GFlowNet for Solving Vehicle Routing Problems"
_ICLR.cc/2026/Conference — Submitted to ICLR 2026_

### Official Review · Reviewer_TAwx · 2025-10-31

**Soundness:** 3
**Presentation:** 3
**Contribution:** 3
**Rating:** 6
**Confidence:** 3

**Summary:**

This paper proposes Diffusion-Enhanced GFlowNet (DEG) to solve vehicle routing problems. Specifically, its key idea is to integrate GFlowNet with diffusion models to facilitate flow expansion in high-reward regions. Meanwhile, it introduces a simple but effective scale adapter for instances to achieve better generalization performance on scales. Experiments on synthetic data and real-world dataset (TSPLib and CVRPLib) demonstrate that DEG can achieve competitive performance over previous methods, especially on large-scale instances.

**Strengths:**

1.	The integration of GFlowNet with diffusion models seems novel. The idea of utilizing the exploration ability of diffusion models to facilitate effective flow expansion for GFlowNet seems to be technically sound.
2.	The proposed method DEG demonstrates good performance, especially on large-scale instances. The ablation studies are clear and sufficient.

**Weaknesses:**

1.	As discussed in the conclusion section of this paper, the performance of DEG on small-scale still fall short of AR methods like LEHD, which limits its practical application where the instance scale can be various.
2.	While the empirical results are good, a more rigorous theoretical analysis of how the diffusion process facilitates exploration, and why edge-specific signals outperform trajectory-level signals would strengthen the work a lot.

**Questions:**

Is the proposed graph-scale adapter available for other methods to enhance their performance? Can you provide some discussions on it?

---

> ### Author Response · Authors · 2025-11-23
>
> We sincerely thank the reviewer for the positive and encouraging evaluation. We appreciate the reviewer’s recognition of the novelty of integrating GFlowNet with diffusion models and the technical soundness of leveraging diffusion-based exploration to enhance flow expansion. We are also grateful for the acknowledgment of DEG’s strong performance and the clarity of our ablation studies. We greatly value the reviewer’s feedback, and our responses to the raised questions and suggestions are provided below.
>
> #### **W1. Comparison with AR**
>
> We thank the reviewer for the careful reading and for highlighting this important point. We agree that, as also discussed in our conclusion, the performance gap between NAR and AR methods such as LEHD，on small-scale instances reflects a well-known structural difference between the two paradigms. While NAR models offer clear advantages in inference efficiency and scalability, they typically lag behind AR models in fine-grained route construction. Our work aims precisely to narrow this gap. As shown in **Table 2** in the paper, DEG already **surpasses the AR state-of-the-art LEHD on large-scale** problems, while also outperforming strong diffusion-based and GFlowNet-based NAR baselines including **DIFUSCO, T2T, FastT2T, DEITSP, GFACS, and AGFN**. We believe these results demonstrate that DEG represents a meaningful step forward for NAR methods and helps push them closer to AR performance while preserving the inherent efficiency and scalability benefits of the NAR paradigm.
>
> #### **W2.1 Exploration**
>
> We appreciate the reviewer’s insightful comment. Regarding how the diffusion process facilitates exploration, the manuscript includes a theoretical analysis in **Lines 215–238**, where we explain how noise injection enables DEG to emphasize exploitation in low-reward regions while promoting exploration in high-reward regions. We will further clarify this part in the revised version to make the theoretical role of diffusion in exploration more explicit to readers. Motivated by the reviewer’s suggestion, we additionally **conducted an experiment to validate this analysis**. The results clearly show the expected transition pattern: early-stage exploitation followed by later-stage exploration, which directly supports the reasoning described in Lines 216–238. As the rebuttal interface does not support figures, we kindly refer the reviewer to the updated  **Section A.1 and corresponding Figure 2 of Appendix** in the revised paper for a visual illustration of this behavior.
> #### **W2.2 Edge-specific Signals**
>
> We thank the reviewer for raising this important point. In our formulation, edge-specific signals offer advantages over trajectory-level signals because **trajectory-level rewards inherently mix the contributions of all edges along a sampled path**. If a trajectory contains even a single poorly performing edge, its aggregated trajectory-level reward becomes dominated by this unfavorable component. Consequently, when this trajectory-level signal is propagated back through the forward policy $P_F$, all edges on the trajectory, including those that were actually good, receive a degraded training signal. This effect can impair credit assignment and introduce instability, particularly on longer trajectories where the variance of aggregated rewards is higher.
>
> In contrast, DEG computes **edge-specific signals** by averaging reward contributions over many trajectories, which **effectively reduces the influence of isolated bad edges and yields a cleaner, low-variance estimate for each edge**. This promotes more stable credit propagation and allows good edges to be reinforced even when they occasionally appear in suboptimal trajectories. In practice, this leads to faster convergence and improved learning stability.
>
> To further support this argument, we provide empirical learning-efficiency comparisons **Table h.1 and Table h.2**. The results show that DEG achieves better learning efficiency than the vanilla GFlowNet under identical training budgets, confirming that edge-specific signals indeed yield more efficient and stable learning dynamics. We have provided these results in the revised paper at **Section 5.4 of the main paper and Table 9 in the Appendix**.
>
> ##### Table h.1 Comparison of Training Efficiency on TSP
>
> | TSP               | 200(Obj.) | Gap(%)    | 500(Obj.) | Gap(%)    | 1000(Obj.) | Gap(%)    |
> | -| -- |-- |- |- |- |-|
> | original GFlowNet | 12.08     | 13.75     | 19.02     | 14.92     | 27.29      | 17.93     |
> | DEG               | **11.91** | **12.15** | **18.79** | **13.53** | **26.80**  | **15.82** |
>
> ##### Table h.2 Comparison of Training Efficiency on CVRP
>
> | CVRP              | 200(Obj.) | Gap(%)    | 500(Obj.) | Gap(%)    | 1000(Obj.) | Gap(%)   |
> |-- | -- | ---| --| ---| ----| -|
> | original GFlowNet | 32.03     | 14.23     | 71.49     | 13.22     | 132.47     | 10.84    |
> | DEG               | **31.90** | **13.77** | **70.15** | **11.10** | **131.23** | **9.80** |

---

> > ### Author Response · Authors · 2025-11-23
> >
> > #### **Q1. GSA Applicability**
> >
> > We thank the reviewer for this great suggestion. Following the reviewer’s recommendation, we applied GSA to two strong frameworks, **GFACS** and **AGFN**, to examine whether similar scale-adaptation benefits can be observed. As shown in **Table i.1 and Table i.2**, incorporating GSA **consistently improves the generalization performance of both models** when evaluated on scales different from their training sizes. These results confirm that GSA is a reusable and broadly beneficial component rather than a DEG-specific mechanism, and that DEG still outperforms these methods even when all of them are equipped with GSA. We have added the corresponding results to the revised manuscript, including updates to the **Section B.6 and Table 13-14 in the Appendix.**
> >
> > ##### Table i.1. GSA Application to GFACS and AGFN on TSP
> >
> > |                | 500       | Gap(%)    | 1000      | Gap(%)    | 3000      | Gap(%)    | 5000      | Gap(%)    | 8000       | Gap(%)    | 10000      | Gap(%)    |
> > | -------------- | --------- | --------- | --------- | --------- | --------- | --------- | --------- | --------- | ---------- | --------- | ---------- | --------- |
> > |                | 16.55     | -         | 23.14     | -         | 40.31     | -         | 51.84     | -         | 65.89      | -         | 75.78      | -         |
> > | GFACS          | 22.97     | 38.79     | 38.58     | 66.72     | 69.63     | 72.74     | 97.93     | 88.91     | 120.76     | 83.28     | 140.84     | 85.85     |
> > | GFACS with GSA | **22.10** | **33.53** | **35.90** | **55.14** | **62.70** | **55.54** | **88.46** | **70.64** | **113.88** | **72.07** | **129.39** | **70.74** |
> > | AGFN           | 19.04     | 15.05     | 27.10     | 17.11     | 55.37     | 37.36     | 80.93     | 56.12     | 118.53     | 79.89     | 143.85     | 89.83     |
> > | AGFN with GSA  | **18.87** | **14.02** | **26.92** | **16.34** | **53.26** | **32.13** | **75.95** | **46.51** | **110.52** | **67.73** | **127.76** | **68.59** |
> > | DEG with GSA   | 18.47     | 11.60     | 26.46     | 14.35     | 47.82     | 18.63     | 63.64     | 22.57     | 83.13      | 26.16     | 94.67      | 24.94     |
> >
> > ##### Table i.2. GSA Application to GFACS and AGFN on CVRP
> > *Note: The gap (%) is computed with respect to LKH(10,000) for CVRP instances with fewer than 1,000 nodes, and with respect to LKH(1,000) for instances above 3,000 nodes due to time limits, following the same experimental settings as in the original paper.*
> >
> > |                | 500       | Gap(%)    | 1000       | Gap(%)    | 3000       | Gap(%)    | 5000       | Gap(%)    | 8000       | Gap(%) | 10000       | Gap(%) |
> > | -------------- | --------- | --------- | ---------- | --------- | ---------- | --------- | ---------- | --------- | ---------- | ------ | ----------- | ------ |
> > |                | 63.14     | -         | 119.52     | -         | 380.26     | -         | 642.95     | -         | -          | -      | -           | -      |
> > | GFACS          | 80.77     | 27.92     | 158.97     | 33.01     | 393.27     | 3.42      | 650.62     | 1.19      | 1038.30    | -      | 1439.19     | -      |
> > | GFACS with GSA | **80.26** | **27.11** | **146.76** | **22.79** | **383.50** | **0.85**  | **622.67** | **-3.15** | **922.24** | **-**  | **1323.82** | **-**  |
> > | AGFN           | 71.05     | 12.53     | 133.97     | 12.09     | 368.97     | -2.97     | 608.58     | -5.35     | 917.24     | -      | 1140.46     | -      |
> > | AGFN with GSA  | **70.74** | **12.04** | **131.60** | **10.11** | **366.72** | **-3.56** | **602.76** | **-6.25** | **909.15** | **-**  | **1131.58** | **-**  |
> > | DEG with GSA   | 69.76     | 10.48     | 129.67     | 8.49      | 362.11     | -4.77     | 596.43     | -7.24     | 907.65     | -      | 1125.04     | -      |
> >
> > We are truly grateful for the reviewer’s detailed and constructive feedback. Your comments have significantly strengthened the presentation and scope of our work. If you have any further questions or suggestions, we would be more than happy to address them. Thank you again for your valuable insights.

---

### Official Review · Reviewer_bkKu · 2025-11-01

**Soundness:** 3
**Presentation:** 2
**Contribution:** 3
**Rating:** 6
**Confidence:** 3

**Summary:**

The paper proposes Diffusion-Enhanced GFlowNet (DEG), a framework that injects a diffusion process into GFlowNet training to encourage flow expansion toward high-reward regions for TSP/CVRP. Concretely, the method constructs edge-specific backward signals by aggregating trajectory rewards, normalizes them (z-score + Gaussian CDF), perturbs them with a diffusion noise schedule and then denoises within the GFlowNet paradigm by using the noised backward policy in the trajectory balance loss. The authors also introduce a Graph-Scale Adapter (GSA) to align spatial statistics across train/test graph sizes, claiming improved scalability up to 10k nodes. Empirically, DEG reports consistent improvements over GFlowNet- and diffusion-based baselines on synthetic TSP/CVRP and shows favorable gaps on TSPLib/CVRPLib-style instances, often with strong runtime.

**Strengths:**

1. **Methodological clarity**. The core mechanism—edge-level reward estimation (Eq. 3–5), diffusion corruption (Eq. 6), and its integration into TB training (Eq. 10–12)—is described with reasonable detail, and the learning-target interpretation is intuitively argued.

2. **Originality**. Using GFlowNet-induced edge-specific signals as the substrate for diffusion in VRP appears new and interesting; most prior diffusion solvers are TSP-only and supervised, whereas DEG claims a more general, reward-driven integration.

3. **Empirics**. Tables indicate strong results across sizes, including large-scale TSP up to 10k nodes with favorable runtime, and competitive CVRP gaps vs LEHD/POMO and GFlowNet baselines. However, some comparisons (e.g., CVRPLib subsets and enormous gaps for some baselines) merit closer scrutiny and clarified protocols (datasets, post-processing, hardware).

**Weaknesses:**

1. **Lack of empirical, quantitative evidence for the proposed method**. The paper's central and most interesting claim is that the diffusion-based noise schedule creates a natural learning curriculum, shifting the learning objective from exploitation (proportional to R(τ)²) to exploration (proportional to R(τ)). This idea, however, hinges on a critical assumption presented in lines 222-224: "Early in training... the noised backward policy ... closely approximates the reward distribution." While this assumption is intuitive, the paper does not provide any empirical or quantitative evidence to validate it. The entire justification for using a diffusion model rests on this premise, but it is left as a claim rather than a verified fact. To significantly strengthen the paper's contribution, the authors should provide evidence to support this.

2. **Single-scale training**.  All models are trained only on 100-node synthetic instances, and scalability to larger graphs is achieved mainly via GSA rescaling at test time. This makes it unclear whether the method itself is scale-robust, or whether the rescaling step is doing most of the work. The authors should show results where models are also trained on other node sizes and evaluated with/without rescaling, to demonstrate that performance is not tied to the 100-node training setup.

**Questions:**

1. Could you provide empirical or quantitative evidence to support the key assumption stated in lines 222–224 that “early in training, the noised backward policy closely approximates the reward distribution”?

2. What is the computational and memory overhead of maintaining and denoising the edge-level backward maps compared to a vanilla GFlowNet? Is the added complexity justified by the performance gain across all problem sizes, or only for large graphs?

3. Since GSA addresses training–inference scale mismatch and not a DEG-specific issue, it seems reusable. Can you show its impact when applied to at least other VRP baselines?

---

> ### Author Response · Authors · 2025-11-23
>
> We sincerely thank the reviewer for the positive and encouraging evaluation. We appreciate the reviewer’s acknowledgement of the methodological clarity, originality, and empirical strength of our work. We also greatly value the reviewer’s constructive comments regarding experimental protocols and comparison details. These suggestions are very helpful, and we have carefully addressed each point in our responses below.
>
> #### **CVRPLib**
> We thank the reviewer for this thoughtful observation. The performance gaps observed on certain CVRPLib subsets are indeed **common** in the VRP literature, especially for **neural-network–based methods**. This discrepancy arises because learning-based solvers are trained on synthetic uniform distributions, whereas CVRPLib contains highly heterogeneous, real-world–like instances with irregular spatial structures, clustered demands, and non-uniform route patterns. When models trained purely on uniform synthetic data are evaluated on these more complex distributions, larger performance gaps naturally occur for some instances.
>
> This phenomenon has been widely reported in prior neural VRP papers. Nonetheless, DEG still achieves competitive or superior performance on many CVRPLib subsets, demonstrating strong generalization despite the distribution shift.
>
> #### **W1, Q1. Empirical Evidence**
>
> We thank the reviewer for this insightful suggestion, which has been very helpful in strengthening our paper. We provide empirical evidence in the paper supporting the assumption stated in Lines 222–224, and the intent the trajectories sampled by DEG during training exhibit the expected behavior aligned with the target distribution: they transition from an early exploitation phase to a later exploration phase, corresponding to the shift from Eq. (8) to Eq. (9). As the rebuttal interface does not support figures, we kindly refer the reviewer to the updated  **Section A.1 and corresponding Figure 2 of Appendix** in the revised paper for a visual illustration of this behavior.
>
> For the assumption stated in lines 222–224, as shown in Figure 2, where $R^2$ denotes the coefficient of determination, values closer to 1 indicate a stronger goodness of fit; in practice, an $R^2$ above 0.9 is typically regarded as strong evidence of a good linear relationship. This empirical observation confirms the validity of Eq. (13):
>
> $P_F(\tau)\;\propto\;R(\tau)\Bigl(\sqrt{\alpha_t}\, R(\tau)+\sqrt{1-\alpha_t}\,\epsilon\Bigr),$
>
> which characterizes the diffusion-modulated forward policy during training.
>
> In the early stages of training, the diffusion schedule satisfies $\sqrt{\alpha_t}\approx 1$, and Eq.~(13) therefore reduces to Eq. (8):
> $P_F(\tau) \propto R(\tau)^2,$
> demonstrating that DEG initially operates in an exploitation regime dominated by squared-reward weighting before gradually transitioning toward enhanced exploration in high-reward regions.
> #### **W2. Single-scale training**
>
> We appreciate the reviewer for highlighting this point, as adding such experiments indeed provides a more complete demonstration of the scale-robustness of our framework. Following the reviewer’s recommendation, we conducted additional experiments in which DEG is **trained on multiple node sizes**. Specifically, for both TSP and CVRP, we trained models on instances of size **100, 200, 500, and 1000**, and evaluated them on larger test sets with **500, 1000, 3000, 5000, 8000, and 10000 nodes**. For each scale, we report results **with and without GSA rescaling** to explicitly disentangle the contributions of the model and the scaling mechanism.
>
> The new results (shown in **Table g.1, Table g.2, Table g.3, and Table g.4**) demonstrate that DEG maintains strong performance even **without** GSA, while the use of GSA further improves generalization on very large-scale instances. These findings confirm that the scalability of DEG is **not** tied to training on 100-node instances alone and that the model itself exhibits substantial inherent scale-robustness. We have incorporated all corresponding results into the revised manuscript, including updates to the **Section B.5 and Tables 11 and 12 in the Appendix**.

---

> > ### Author Response · Authors · 2025-11-23
> >
> > ##### Table g.1. Training on Different Scales for TSP and Testing without GSA
> >
> > *Note: The values along the vertical axis denote the instance sizes used during training, while the values along the horizontal axis correspond to the instance sizes used for testing. "--" indicates that the model was unable to produce a result for a given instance either within the 90-minute time limit or due to GPU memory constraints.*
> >
> > |      | 500(Obj.) | Gap(%)   | 1000(Obj.) | Gap(%)    | 3000(Obj.) | Gap(%)    | 5000(Obj.) | Gap(%)    | 8000(Obj.) | Gap(%) | 10000(Obj.) | Gap(%) |
> > | ---- | --------- | -------- | ---------- | --------- | ---------- | --------- | ---------- | --------- | ---------- | ------ | ----------- | ------ |
> > | 100  | 18.54     | 12.02    | 26.74      | 15.56     | 48.45      | 20.19     | 64.97      | 25.33     | -          | -      | -           | -      |
> > | 200  | 18.44     | 11.42    | 26.50      | 14.52     | 47.59      | 18.06     | 64.16      | 23.77     | -          | -      | -           | -      |
> > | 500  | **18.05** | **9.06** | 26.42      | 14.17     | 47.19      | 17.07     | **61.55**  | **18.73** | -          | -      | -           | -      |
> > | 1000 | 18.41     | 11.24    | **25.87**  | **11.80** | **46.32**  | **14.91** | 63.14      | 21.80     | -          | -      | -           | -      |
> >
> > ##### Table g.2. Training on Different Scales for TSP and Testing with GSA
> >
> > |      | 500(Obj.) | Gap(%)   | 1000(Obj.) | Gap(%)    | 3000(Obj.) | Gap(%)    | 5000(Obj.) | Gap(%)    | 8000(Obj.) | Gap(%)    | 10000(Obj.) | Gap(%)    |
> > | ---- | --------- | -------- | ---------- | --------- | ---------- | --------- | ---------- | --------- | ---------- | --------- | ----------- | --------- |
> > | 100  | 18.47     | 11.60    | 26.46      | 14.35     | 47.82      | 18.63     | 63.64      | 22.57     | 83.13      | 26.16     | 94.67       | 24.94     |
> > | 200  | 18.33     | 10.76    | 26.34      | 13.83     | 46.91      | 16.37     | 62.16      | 19.91     | 82.19      | 24.74     | 91.09       | 20.20     |
> > | 500  | **18.08** | **9.24** | 25.97      | 12.23     | 46.05      | **14.24**     | **61.11**  | **17.88** | **78.16**  | **18.62** | **87.44**   | **15.39** |
> > | 1000 | 18.28     | 10.45    | **25.87**  | **11.80** | **45.09**  | 14.86 | 62.31      | 20.20     | 79.06      | 19.99     | 89.58       | 18.21     |
> >
> > ##### Table g.3. Training on Different Scales for CVRP and Testing without GSA
> > *Note: The gap (%) is computed with respect to LKH(10,000) for CVRP instances with fewer than 1,000 nodes, and with respect to LKH(1,000) for instances above 3,000 nodes due to time limits, following the same experimental settings as in the original paper. "--" indicates that the model was unable to produce a result for a given instance either within the 90-minute time limit or due to GPU memory constraints.*
> >
> > |      | 500(Obj.) | Gap(%)   | 1000(Obj.) | Gap(%) | 3000(Obj.) | Gap(%)    | 5000(Obj.) | Gap(%)     | 8000(Obj.) | Gap(%) | 10000(Obj.) | Gap(%) |
> > | ---- | --------- | -------- | ---------- | ------ | ---------- | --------- | ---------- | ---------- | ---------- | ------ | ----------- | ------ |
> > | 100  | 69.76     | 10.48    | 129.67     | 8.49   | 362.11     | -4.77     | 596.43     | -7.24      | 907.65     | -      | 1125.04     | -      |
> > | 200  | 67.75     | 7.30     | 127.70     | 6.84   | 362.43     | -4.69     | 580.73     | -9.68      | 987.01     | -      | 1094.82     | -      |
> > | 500  | **66.76** | **5.73** | 124.70     | 4.34   | 360.80     | -5.12     | 577.24     | -10.22     | 894.98     | -      | 1072.27     | -      |
> > | 1000 | 69.68     | 10.36    | **123.13** | **3.02**   | **360.42** | **-5.21** | **568.73** | **-11.54** | **885.18**     | -      | **1060.53**     | -      |
> >
> > ##### Table g.4. Training on Different Scales for CVRP and Testing with GSA
> >
> > |      | 500(Obj.) | Gap(%)   | 1000(Obj.) | Gap(%)   | 3000(Obj.) | Gap(%)    | 5000(Obj.) | Gap(%) | 8000(Obj.) | Gap(%) | 10000(Obj.) | Gap(%) |
> > | ---- | --------- | -------- | ---------- | -------- | ---------- | --------- | ---------- | ------ | ---------- | ------ | ----------- | ------ |
> > | 100  | 70.86     | 12.23    | 132.65     | 10.74    | 368.99     | -2.96     | -          | -      | -          | -      | -           | -      |
> > | 200  | 68.36     | 8.37     | 129.50     | 8.35     | 368.85     | -3.00     | -          | -      | -          | -      | -           | -      |
> > | 500  | **66.76** | **5.73** | 125.65     | 5.13     | 367.13     | -3.45     | -          | -      | -          | -      | -           | -      |
> > | 1000 | 70.54     | 11.72    | **123.13** | **3.02** | **366.19** | **-3.70** | -          | -      | -          | -      | -           | -      |

---

> > > ### Author Response · Authors · 2025-11-23
> > >
> > > #### **Q2. Computational and Memory Overhead and Performance Gain**
> > >
> > > We thank the reviewer for raising this important question. In DEG, the **overhead** introduced by the edge-level backward map is very small. Although the map is represented as an N×N matrix, its memory footprint is negligible in practice, for example, at N=100, it occupies only about **40 KB**, and even at N=500, it remains roughly **1 MB**, far below typical GPU memory limits. The computational cost is similarly minimal: updating edge-level rewards from sampled trajectories requires only **O(N)** operations per step, and the subsequent noise injection involves simple element-wise operations with negligible additional cost; while this operation promotes more stable credit propagation.
> > >
> > > Empirically, as show in **Table h.1 and Table h.2**, under identical training configurations and training steps, DEG achieves higher optimization quality and better diversity than the vanilla GFlowNet, indicating the **improved learning efficiency**.  We have provided these results in the revised paper at **Section 5.4 of the main paper and Table 9 in the Appendix**.
> > >
> > > Moreover, the added complexity is justified across all problem sizes: as shown in Section 5.4 and Table 5 in the paper, DEG consistently improves performance on TSP and CVRP from 500-node to 10000-node instance, with **strong gains at all sizes**.
> > >
> > > ##### Table h.1 Comparison of Training Efficiency on TSP
> > >
> > > | TSP               | 200(Obj.) | Gap(%)    | 500(Obj.) | Gap(%)    | 1000(Obj.) | Gap(%)    |
> > > | ----------------- | --------- | --------- | --------- | --------- | ---------- | --------- |
> > > | original GFlowNet | 12.08     | 13.75     | 19.02     | 14.92     | 27.29      | 17.93     |
> > > | DEG               | **11.91** | **12.15** | **18.79** | **13.53** | **26.80**  | **15.82** |
> > >
> > > ##### Table h.2 Comparison of Training Efficiency on CVRP
> > >
> > > | CVRP              | 200(Obj.) | Gap(%)    | 500(Obj.) | Gap(%)    | 1000(Obj.) | Gap(%)   |
> > > | ----------------- | --------- | --------- | --------- | --------- | ---------- | -------- |
> > > | original GFlowNet | 32.03     | 14.23     | 71.49     | 13.22     | 132.47     | 10.84    |
> > > | DEG               | **31.90** | **13.77** | **70.15** | **11.10** | **131.23** | **9.80** |
> > > #### **Q3. GSA Applicability**
> > >
> > > We thank the reviewer for this great suggestion. Following the reviewer’s recommendation, we applied GSA to two strong frameworks, **GFACS** and **AGFN**, to examine whether similar scale-adaptation benefits can be observed. As shown in **Table i.1 and Table i.2**, incorporating GSA **consistently improves the generalization performance of both models** when evaluated on scales different from their training sizes. These results confirm that GSA is a reusable and broadly beneficial component rather than a DEG-specific mechanism, and that DEG still outperforms these methods even when all of them are equipped with GSA. We have added the corresponding results to the revised manuscript, including updates to the **Section B.6 and Table 13-14 in the Appendix.**
> > >
> > > ##### Table i.1. GSA Application to GFACS and AGFN on TSP
> > >
> > > |                | 500       | Gap(%)    | 1000      | Gap(%)    | 3000      | Gap(%)    | 5000      | Gap(%)    | 8000       | Gap(%)    | 10000      | Gap(%)    |
> > > | -------------- | --------- | --------- | --------- | --------- | --------- | --------- | --------- | --------- | ---------- | --------- | ---------- | --------- |
> > > | GFACS          | 22.97     | 38.79     | 38.58     | 66.72     | 69.63     | 72.74     | 97.93     | 88.91     | 120.76     | 83.28     | 140.84     | 85.85     |
> > > | GFACS with GSA | **22.10** | **33.53** | **35.90** | **55.14** | **62.70** | **55.54** | **88.46** | **70.64** | **113.88** | **72.07** | **129.39** | **70.74** |
> > > | AGFN           | 19.04     | 15.05     | 27.10     | 17.11     | 55.37     | 37.36     | 80.93     | 56.12     | 118.53     | 79.89     | 143.85     | 89.83     |
> > > | AGFN with GSA  | **18.87** | **14.02** | **26.92** | **16.34** | **53.26** | **32.13** | **75.95** | **46.51** | **110.52** | **67.73** | **127.76** | **68.59** |
> > > | DEG with GSA   | 18.47     | 11.60     | 26.46     | 14.35     | 47.82     | 18.63     | 63.64     | 22.57     | 83.13      | 26.16     | 94.67      | 24.94     |

---

> > > > ### Author Response · Authors · 2025-11-23
> > > >
> > > > ##### Table i.2. GSA Application to GFACS and AGFN on CVRP
> > > > *Note: The gap (%) is computed with respect to LKH(10,000) for CVRP instances with fewer than 1,000 nodes, and with respect to LKH(1,000) for instances above 3,000 nodes due to time limits, following the same experimental settings as in the original paper.*
> > > >
> > > > |                | 500       | Gap(%)    | 1000       | Gap(%)    | 3000       | Gap(%)    | 5000       | Gap(%)    | 8000       | Gap(%) | 10000       | Gap(%) |
> > > > | -------------- | --------- | --------- | ---------- | --------- | ---------- | --------- | ---------- | --------- | ---------- | ------ | ----------- | ------ |
> > > > | GFACS          | 80.77     | 27.92     | 158.97     | 33.01     | 393.27     | 3.42      | 650.62     | 1.19      | 1038.30    | -      | 1439.19     | -      |
> > > > | GFACS with GSA | **80.26** | **27.11** | **146.76** | **22.79** | **383.50** | **0.85**  | **622.67** | **-3.15** | **922.24** | **-**  | **1323.82** | **-**  |
> > > > | AGFN           | 71.05     | 12.53     | 133.97     | 12.09     | 368.97     | -2.97     | 608.58     | -5.35     | 917.24     | -      | 1140.46     | -      |
> > > > | AGFN with GSA  | **70.74** | **12.04** | **131.60** | **10.11** | **366.72** | **-3.56** | **602.76** | **-6.25** | **909.15** | **-**  | **1131.58** | **-**  |
> > > > | DEG with GSA   | 69.76     | 10.48     | 129.67     | 8.49      | 362.11     | -4.77     | 596.43     | -7.24     | 907.65     | -      | 1125.04     | -      |
> > > >
> > > >
> > > >
> > > > We sincerely appreciate the reviewer’s thoughtful and constructive feedback. Your suggestions have been extremely valuable and have significantly improved the clarity, completeness, and overall quality of our work. If you have any additional questions or recommendations, we would be more than happy to address them. Thank you again for your time and insightful input.

---

### Official Review · Reviewer_QC6P · 2025-11-01

**Soundness:** 3
**Presentation:** 2
**Contribution:** 2
**Rating:** 6
**Confidence:** 3

**Summary:**

This paper introduces Diffusion-Enhanced GFlowNet (DEG), a novel framework that synergistically integrates GFlowNet with diffusion models to address the challenge of insufficient flow expansion in high-reward regions for solving Vehicle Routing Problems (VRPs). The core innovation lies in using GFlowNet to generate edge-specific backward signals, which are then perturbed and denoised via a diffusion process to guide richer exploration. The proposed method, complemented by a Graph-Scale Adapter for enhanced scalability.

**Strengths:**

1.	The idea of injecting a diffusion-like noise mechanism into GFlowNet training is novel and interesting. It effectively enhances the model's exploration ability within high-reward regions, addressing a key limitation of existing GFlowNet-based solvers.

2.	Extensive experiments on large-scale TSP and CVRP datasets convincingly demonstrate the superior performance and strong generalization ability of DEG. The results on very large CVRP instances (N ≥ 5000) are particularly impressive, showing that the method can even outperform the well-known LKH solver.

**Weaknesses:**

1.	The experiments lack a comprehensive evaluation on small-scale problems (N ≤ 500). While tackling large-scale instances is a crucial goal for neural solvers, demonstrating robust performance across diverse problem scales is also important, especially for a method that aims to improve general training framework.

2.	The application domain is currently limited to routing problems (TSP and CVRP). Given that DEG appears to be a general training framework for neural combinatorial optimization, evaluating its effectiveness on other related problems (e.g., scheduling, maximum independent set) would significantly broaden the paper's impact.

**Questions:**

The presented experiments primarily focus on generalization performance. I am curious about the fundamental optimization ability of DEG compared to the original GFlowNet method, standard Reinforcement Learning (RL) methods, and newly developed preference optimization methods [1]. To provide a clearer comparison of learning efficiency, could the authors conduct experiments on a fixed problem scale and distribution while aligning the number of training instances/samples across all compared methods?

[1] Preference Optimization for Combinatorial Optimization Problems. ICML 2025.

---

> ### Author Response · Authors · 2025-11-23
>
> We sincerely thank the reviewer for the positive and encouraging feedback. We greatly appreciate the recognition of our contribution, particularly the novelty of introducing a diffusion-like noise mechanism into GFlowNet training and the strong empirical performance of DEG on large-scale TSP and CVRP tasks. The reviewer’s constructive comments are highly valuable, and we have carefully addressed each of them. Our detailed responses are provided below.
>
> #### **W1. Small-Scale Problems**
>
> We thank the reviewer for highlighting the importance of evaluating performance on small-scale problems. We agree that demonstrating robustness across different instance sizes is valuable. Based this suggestion, we have added additional experiments on small-scale TSP and CVRP instances, including 100 node and 200 node. These evaluations include comparisons with both diffusion-based and GFlowNet-based baselines. Since diffusion-based methods cannot directly handle CVRP, we report only the GFlowNet-based baselines for that task. The results are shown in **Table d.1 and Table d.2**.  As shown, our model DEG surpasses all diffusion-based and GFlowNet-based methods on TSP200, CVRP100, and CVRP200, and outperforms nearly all baselines on TSP100. The new results have been incorporated into the **Section B.4 and Table 8 at Appendix** in the revised manuscript.
>
> ##### Table d.1. Comparison on TSP instances with 100 and 200 nodes
>
> | TSP                         | 100(Obj.)     | Time(s) | Gap(%)   | 200(Obj.)       | Time(s) | Gap(%)   |
> | --------------------------- | -------- | ------- | -------- | --------- | ------- | -------- |
> | LKH                         | 7.76     | 1.50    | -        | 10.62     | 9.80    | -        |
> | GFACS                       | 8.80     | 0.53    | 13.54    | 13.20     | 1.15    | 24.29    |
> | AGFN                        | 8.49     | 0.05    | 9.55     | 11.82     | 0.08    | 11.30    |
> | DEG                         | **8.40** | 0.06    | **8.25** | **11.65** | 0.10    | **9.42** |
> | DIFUSCO(with local search)  | 7.88     | 0.42    | 1.68     | 10.80     | 1.53    | 1.70     |
> | T2T(with local search)      | 7.80     | 0.37    | 0.65     | 10.80     | 1.15    | 1.72     |
> | DEITSP(with local search)   | **7.78** | 0.08    | **0.39** | 10.76     | 1.20    | 1.32     |
> | Fast T2T(with local search) | 7.82     | 0.11    | 0.90     | 10.78     | 1.19    | 1.51     |
> | DEG(with local search)      | 7.79     | 0.07    | 0.52     | **10.76** | 1.15    | **1.32** |
>
> ##### Table d.2. Comparison on CVRP instances with 100 and 200 nodes
>
> |CVRP|100(Obj.)|Time(s)|Gap(%)|200(Obj.)|Time(s)|Gap(%) |
> |-|-|-|-|-|-|-|
> | LKH   | 15.57     | 18.80   | -        | 28.04     | 50.83   | -|
> | GFACS | 19.26     | 1.90    | 23.70    | 35.36     | 3.14    | 26.11     |
> | AGFN  | 17.78     | 0.06    | 14.19    | 31.60     | 0.16    | 12.70     |
> | DEG   | **17.07** | 0.07    | **9.63** | **31.00** | 0.18    | **10.56** |
> #### **W2. Applicability to Other Combinatorial Optimization Problems**
>
> We thank the reviewer for the great suggestion. While this paper primarily focuses on routing problems as indicated by the title, we fully agree that evaluating DEG on a broader range of combinatorial optimization tasks would further broaden the impact of the proposed training framework. Motivated by the reviewer’s comment, we extended our experiments to additional problem domains.
>
> Specifically, we considered: (1) **SMTWTP (Single-Machine Total Weighted Tardiness Problem)** is a classical single-machine scheduling problem where the objective is to minimize the total weighted tardiness of jobs processed on one machine. In **Table e.1**, the instance sizes 100, 200, and 500 denote the number of jobs in the scheduling instance. The reported Obj. values correspond to the objective function value of SMTWTP, i.e., the total weighted tardiness achieved by each method. (2) **BPP (Bin Packing Problem)** is a packing optimization problem that aims to minimize the number of bins required to accommodate items of varying sizes under capacity constraints, thereby improving space utilization. In **Table e.2**, the instance sizes 120, 250, and 500 indicate the number of items in each bin-packing instance. The Obj. (%) values represent the space utilization rate, i.e., the proportion of total bin capacity effectively used by the packed items, where higher percentages indicate more efficient utilization of available space.
>
> The results, summarized in **Table e.1 and Table e.2**, show that DEG outperforms both **DeepACO**[1] and **GFACS** on these tasks, further demonstrating its potential as a general neural combinatorial optimization framework beyond routing. We greatly appreciate the reviewer’s suggestion, which has meaningfully broadened the scope and impact of our paper. These results have been added to **Section 5.5 of the main paper and Table 10 in the Appendix**.
>
> [1] DeepACO: Neural-enhanced ant systems for combinatorial optimization. NeurIPS 2023.

---

> ### Author Response · Authors · 2025-11-23
>
> ##### Table e.1 Results on SMTWTP
>
> |         | 100(Obj.)$\downarrow$ | Time(s)  | 200(Obj.)$\downarrow$  | Time(s)  | 500(Obj.)$\downarrow$  | Time(s)  |
> | ------- | --------- | -------- | ---------- | -------- | ---------- | -------- |
> | ACO     | 361.22    | 0.82     | 1608.26    | 1.70     | 11015.39   | 2.84     |
> | DeepACO | 223.36    | 0.93     | 905.19     | 1.81     | 1567.20    | 3.56     |
> | GFACS   | 40.67     | 1.04     | 119.32     | 2.25     | 559.31     | 3.68     |
> | DEG     | **37.70** | **0.14** | **103.13** | **0.34** | **510.67** | **0.98** |
>
> ##### Table e.2 Results on BPP
>
> | BPP     | 120(Obj.(%))$\uparrow$  | Time(s)  | 250(Obj.(%))$\uparrow$  | Time(s)  | 500(Obj.(%))$\uparrow$  | Time(s)  |
> | ------- | ------------ | -------- | ------------ | -------- | ------------ | -------- |
> | ACO     | 90.65        | 2.95     | 90.36        | 5.85     | 90.17        | 10.00    |
> | DeepACO | 93.46        | 2.37     | 93.10        | 5.50     | 93.08        | 10.75    |
> | GFACS   | 94.68        | 2.38     | 94.03        | 5.56     | 93.65        | 10.77    |
> | DEG     | **96.94**    | **0.27** | **96.30**    | **0.58** | **95.37**    | **1.38** |
> #### **Q1. Learning Efficiency**
>
> We thank the reviewer for this valuable suggestion. Evaluating learning efficiency is indeed an important perspective for understanding the fundamental optimization ability of different learning paradigms. We also appreciate the pointer to the preference optimization (PO) method [1], which is a strong and relevant baseline for this comparison.
>
> Following the reviewer’s suggestion, we conducted additional experiments on a fixed problem scale and distribution. Specifically, we used **10,000 uniformly sampled TSP and CVRP instances at size n=200** and trained all models under identical configurations, differing only in the training paradigm (RL, standard GFlowNet, PO [1], and our DEG). We then evaluated the trained models on **TSP/CVRP test instances at n=200, n=500, and n=1000**. The results are presented in **Table f.1 and Table f.2**, and they show that DEG demonstrates strong optimization ability and efficient learning compared to these baselines under controlled and comparable training conditions. We have provided these results in the revised paper at **Section 5.4 of the main paper and Table 9 in the Appendix**.
>
> Again, we sincerely thank the reviewer for bringing the PO work [2] to our attention. It is actually an excellent and theoretically insightful contribution, and has sparked our interest in exploring deeper connections between preference optimization and GFlowNets. We believe that establishing such theoretical links could be a promising direction for future research and may further enhance the strengths of both frameworks.
>
> [1] Preference Optimization for Combinatorial Optimization Problems. ICML 2025.
> ##### Table f.1 Comparison of Training Efficiency on TSP
>
> | TSP                    | 200(Obj.) | Gap(%)    | 500(Obj.) | Gap(%)    | 1000(Obj.) | Gap(%)    |
> | ---------------------- | --------- | --------- | --------- | --------- | ---------- | --------- |
> | Reinforcement Learning | 12.12     | 14.12     | 19.28     | 16.50     | 27.82      | 20.22     |
> | original GFlowNet      | 12.08     | 13.75     | 19.02     | 14.92     | 27.29      | 17.93     |
> | PO                     | 11.97     | 12.71     | 18.93     | 14.38     | 27.11      | 17.16     |
> | DEG                    | **11.91** | **12.15** | **18.79** | **13.53** | **26.80**  | **15.82** |
> ##### Table f.2 Comparison of Training Efficiency on CVRP
>
> | CVRP                 | 200(Obj.) | Gap(%)    | 500(Obj.) | Gap(%)    | 1000(Obj.) | Gap(%)   |
> | ---------------------- | --------- | --------- | --------- | --------- | ---------- | -------- |
> | Reinforcement Learning | 32.94     | 17.48     | 72.50     | 14.82     | 133.41     | 11.62    |
> | original GFlowNet      | 32.03     | 14.23     | 71.49     | 13.22     | 132.47     | 10.84    |
> | PO                     | 32.13     | 14.59     | 71.77     | 13.67     | 133.39     | 11.60    |
> | DEG                    | **31.90** | **13.77** | **70.15** | **11.10** | **131.23** | **9.80** |
>
> We sincerely thank you for your thoughtful and constructive feedback. Your comments are valuable and have significantly strengthened the quality and clarity of our paper. If you have any further suggestions or questions, we would be very glad to address them.

---

> > ### Comment · Reviewer_QC6P · 2025-11-27
> >
> > Thank you for your response. The additional experiments have addressed most of my concerns. Although the performance of DEG on CVRP100 is weak (9.63% gap), it seems that all the GFlowNet-based methods have this issue. Therefore, I will maintain my positive score but increase my confidence.

---

> > > ### Author Response · Authors · 2025-11-27
> > >
> > > Thank you very much for acknowledging our rebuttal and for maintaining a positive score. While our method outperforms many baselines on large-scale instances, we acknowledge that GFlowNet-based approaches may still be relatively inferior on smaller instances, and addressing this limitation will be an important direction for our future work. Again, we truly appreciate your valuable suggestions and support, and we are more than happy to clarify any remaining concerns if there are still questions during the discussion period.

---

### Official Review · Reviewer_9NRj · 2025-11-03

**Soundness:** 2
**Presentation:** 2
**Contribution:** 2
**Rating:** 2
**Confidence:** 3

**Summary:**

The paper introduces Diffusion Enhanced GFlowNet, a combination of GFlowNet with diffusion technique, to encourage exploration of higher reward regions. Although DEG achieves promising results on several dataset, it still lags behind more recent baselines in terms of speed and performance.

**Strengths:**

- The idea of using diffusion to encourage exploration in GFlowNet is interesting.

**Weaknesses:**

**Major issues**

1. The paper is hard to read for people without a solid background in GFlowNet (like me). I struggled to understand how the network is trained and how the solution is constructed.
2. The main motivation is not properly backed by theoretical and empirical evidence. Questions:
- Adding noise to the backward policy changes the distribution. How can we ensure that DEG samples from the target distribution?
- I would like to see more theoretical and empirical evidence for the claim ''DEG improves the diversity''. The second part of section 4.1 only offers a vague intuition on how diffusion might encourage exploration. The experiment also did not illustrate this point.
3. The performance is weak compared to more recent baselines. For example, [1] achieved much stronger results at even faster speed.

**Minor issues**

- Missing citation to original work on diffusion model (Sohl-Dickstein et al., 2015) (line 60).

[1] Learning to Reduce Search Space for Generalizable Neural Routing Solver. Changliang Zhou, Xi Lin, Zhenkun Wang, Qingfu Zhang

**Questions:**

Please address the weaknesses mentioned above.

---

> ### Author Response · Authors · 2025-11-23
>
> #### **W1. How DEG is Trained and How Solutions Are Constructed**
>
> We apologize for the earlier lack of clarity. Since prior GFlowNet-based works such as GFACS and AGFN have already described how GFlowNets are used in the VRP domain, we followed their conventions and did not elaborate further in the original submission. We value your suggestion, and make the training and inference process of DEG more accessible to readers who may not be familiar with GFlowNet, we provide a VRP-oriented and intuitive explanation below, including its intuitive explanation, the core concept of DEG, its training and inference processes.
>
> **An intuitive explanation of GFlowNets**
> A Generative Flow Network (GFlowNet) is a model that learns to generate structured objects by viewing the construction process as a sequence of decisions. Its core idea is to enforce a balance of "flow’" across all possible paths, so that the probability of producing each final object is proportional to its reward. In this sense, GFlowNet is better understood as a training paradigm that specifies how to learn a policy. Due to the flow consistency learning objective, this paradigm naturally leads to diverse solutions instead of collapsing to a single one, which makes GFlowNets particularly suitable for complex combinatorial problems. We also provide a more comprehensive introduction in **Section C.1 of the appendix**.
>
> **DEG learns an edge-level heatmap policy**
>
> DEG follows a non-autoregressive GFlowNet paradigm in which the neural network produces, in a single forward pass, an *edge-level heatmap* over the entire graph. Given the VRP instance, such as node coordinates, customer demands and depot information, the model outputs a score or unnormalized probability for every directed edge. These edge scores (the heatmap) encode the model’s belief about how promising each possible connection is in the final solution. During inference, they guide the construction of the full VRP solution by selecting edges according to the learned heatmap.
>
> **DEG adjusts the policy so that high-reward routes are sampled more often**
>
> Using the GFlowNet training paradigm, the model is optimized so that the probability of sampling a route is proportional to its corresponding reward. This is fundamentally different from reinforcement learning, which aims at locating a single best route; DEG instead learns a distribution over many high-quality routes.
>
> **Diffusion noise guides exploration toward high-reward regions**
>
> DEG modifies the backward-propagated reward signal by injecting diffusion-style noise to modulate the trade-off between exploration and exploitation. Early in training, the noise level is low, enabling exploitation of promising routing structures. As training progresses, the noise level gradually increases, encouraging broader exploration in high-reward regions.
>
> **DEG Training**
> Based on above, now we can discuss about how DEG is trained. In particular, the training paradigm is shown below based on trajectory balance:
>
> $L(T; \theta)= \sum_{k=1}^{m}( \log \frac{Z(G; \theta) P_F(\tau_k; \theta)}{{R(\tau_k)} {P_{B}({\tau}_{k})}} )^2.$
>
> In this equation, $Z(\mathcal{G}; \theta)$ denotes the source flow, computed from the instance $\mathcal{G}$ through the neural network $\theta$. The forward policy $P_{F}(\tau_k; \theta)$ is the product of the heatmap entries corresponding to all edges along trajectory $\tau$. The reward $R(\tau_k)$ is defined in Equation (3) of the main paper. Finally, $P_{B}({\tau}_{k})$ is the backward policy introduced in our work, where we inject diffusion-style noise to dynamically control the exploitation–exploration behavior during training, and then denoises them within the GFlowNet paradigm.

---

> ### Author Response · Authors · 2025-11-23
>
> **Solution Construction (DEG Inference)**
>
> During training or inference, DEG produces a complete VRP solution directly from the learned edge-level heatmap, without relying on step-by-step autoregressive decisions. **(1. Heatmap generation)** Given a VRP instance, the trained model generates, in a single forward pass, an N×N heatmap that assigns a probability or score to every directed edge. This heatmap reflects the model’s learned preference for which edges are likely to appear in high-reward solutions. **(2. Constraint filtering)** To construct the set of routes, the heatmap is first masked using VRP feasibility constraints, such as vehicle capacity and visit requirements, to ensure that only valid edges remain selectable. **(3. Route construction)** A route is then generated by selecting edges according to their heatmap scores (typically by sampling with probabilities proportional to their heatmap values, or by applying a greedy selection rule to choose the valid edge with highest probability). When the selected edges form a cycle or the vehicle capacity is exhausted, the current route is terminated. The same process is then applied to the remaining unvisited customers until every customer has been assigned to exactly one route.
>
> We thank you again for your helpful suggestion, which has significantly improved the clarity of our presentation. To further assist readers who may not be familiar with GFlowNets, we have added the corresponding explanations in **Section C of the Appendix** in the revised manuscript. If there are still any parts that remain unclear, please feel free to let us know, we would be more than happy to provide additional clarification. Your feedback has been invaluable in improving the clarity and accessibility of our work.
> #### **W2.1 Target Distribution**
>
> We thank the reviewer for this suggestion, which has been very helpful in strengthening our paper.
>
> **(Theoretical explanation)** In practical GFlowNet formulations, the **backward policy** $P_B$ is **a free design component** and has been instantiated in multiple ways across prior work. For example, existing approaches parameterize $P_B$ using dynamically updated neural networks or employ various hand-crafted or path-dependent heuristics. Although these choices differ in how $P_B$ influences the intermediate learning dynamics, they all preserve the same reward-defined target distribution, because $P_B$ does not determine the terminal distribution in GFlowNets. Instead, **$P_B$ implicitly affects how trajectories are selected** (which is explicitly affected by $P_F$) during training: whether through a static rule or a dynamically learned strategy, while the reward function $R$ remains the fundamental quantity that defines the sampling objective.
>
> Building on this theoretical principle, our work introduces a specific construction of $P_B$ by injecting a controllable level of noise into the reward signal to form a noise-scheduled backward policy, where the GFlowNet training objective itself remains unchanged which still enforces flow consistency at convergence. This design encourages more thorough exploration within high-reward regions and provides a smooth mechanism for regulating the model’s exploration--exploitation trade-off. Throughout this process, however, **the reward function $R$ continues to define the target distribution, and the role of $P_B$ is solely to influence how trajectories are chosen based on the current $P_F$**, **rather than to alter the reward-matching objective**.
>
> **(Additional empirical evidence)** We provide additional empirical evidence in the paper showing that the trajectories sampled by DEG during training behave exactly as expected from the target distribution. As the rebuttal interface does not support figures, we kindly refer the reviewer to the updated **Section A.1 and corresponding Figure 2 of Appendix** in the revised paper for a visual illustration of this behavior. In both fingure, $R^2$, which denotes the coefficient of determination and measures goodness of fit, reaches 0.91 and 0.90 for two fingures respectively. In practice, an $R^2$ above 0.9 is typically regarded as strong evidence of a good linear relationship. **The fitting toward the lower-left area in the both figures** shows that after training, the DEG's sample **satisfied to $P_F(\tau) \propto R(\tau)$**, thus DEG samples from the target distribution.

---

> > ### Author Response · Authors · 2025-11-23
> >
> > #### **W2.2 Diversity Experiment**
> >
> > We thank the reviewer for this valuable suggestion. We agree that providing stronger theoretical and empirical evidence for the diversity improvement is important. As shown in **Table a**, we have **added additional experiments on both TSP and CVRP** tasks, comparing the diversity of trajectories generated by (i) the **reinforcement learning** baseline, (ii) the  **vanilla GFlowNet** implementation (the backbone of our work), and (iii) our proposed **DEG** model. The diversity metric is computed as the average number of edges in one trajectory that do not appear in another, a standard way to quantify route-level diversity in VRP; in this experiment, all settings are kept identical across models except for the training paradigm. The new results clearly demonstrate that DEG consistently improves diversity over standard GFlowNets. These results have been incorporated into the main text, along with the corresponding **Section 5.4 and Table 6** in the revised paper.
> >
> > ##### Table a. Diversity Comparison
> >
> > | CVRP     | 100$\uparrow$       | 500$\uparrow$        | 1000$\uparrow$       | 3000$\uparrow$        | 5000$\uparrow$        |
> > | -------- | --------- | ---------- | ---------- | ----------- | ----------- |
> > | RL       | 16.57     | 181.36     | 400.71     | 1210.16     | 2372.48     |
> > | GFlowNet | 19.33     | 212.27     | 443.73     | 1526.56     | 2705.91     |
> > | DEG      | **20.54** | **217.02** | **471.68** | **1648.82** | **2909.12** |
> >
> > | TSP      | 100$\uparrow$       | 500$\uparrow$        | 1000$\uparrow$       | 3000$\uparrow$        | 5000$\uparrow$        |
> > | -------- | --------- | ---------- | ---------- | ----------- | ----------- |
> > | RL       | 17.60     | 96.07      | 217.20     | 819.34      | 1516.64     |
> > | GFlowNet | 20.36     | 126.95     | 294.30     | 1074.53     | 1895.23     |
> > | DEG      | **21.79** | **128.97** | **302.63** | **1145.79** | **2082.36** |
> > #### **W3. Comparison with AR Baselines**
> >
> > We sincerely thank the reviewer for bringing this baseline to our attention. We were not previously aware of this work, and appreciate being made aware of its impressive AR-based performance. However, our study **focuses primarily on Non-Autoregressive (NAR) models**, and we only included the most representative Autoregressive (AR) baselines: POMO and LEHD, for reference. In addition, we noticed that L2R [1] has not yet been formally published, and **its implementation details are not fully available**, making it difficult for us to ensure a fair and reproducible evaluation. We also noticed that its experiments use relatively large batch sizes (128 for TSP and 60 for CVRP), while in our paper all models are evaluated with a batch size of 1 to ensure a fair comparison. This further complicates a direct, controlled comparison at this stage. To provide clearer context for this distinction, we further categorize the fundamental differences between AR and NAR approaches as follows:
> >
> > ##### Table b. Comparison of AR and NAR
> >
> > | Aspect             | Autoregressive (AR) Models                                   | Non-Autoregressive (NAR) Models                              |
> > | ------------------ | ------------------------------------------------------------ | ------------------------------------------------------------ |
> > | Generation Process | Sequential, step-by-step inference using a neural network to construct the route. | A single forward pass of the neural network produces a heatmap, from which the solution is derived. |
> > | Performance        | Strong at fine-grained, locally consistent routing           | More challenging to capture detailed local dependencies      |
> > | Efficiency         | Slower inference due to the sequential bottleneck            | Fast inference with efficient global structure modeling      |
> >
> > NAR models offer inherent advantages in inference speed and scalability, which is why NAR research is one of the most active directions in the VRP community. Under this context, our DEG framework **significantly advances the state of the art within the NAR family**. Across TSP and CVRP, DEG outperforms or matches recently published NAR methods, including **GFACS (AISTATS’25), AGFN (ICLR’25), DIFUSCO (NeurIPS’23), T2T (NeurIPS’23), FastT2T (NeurIPS’24), and DEITSP (KDD’25)**. Moreover, DEG achieves performance that is **competitive even with AR-based approaches**, which is an encouraging step forward for the NAR paradigm.

---

> > > ### Author Response · Authors · 2025-11-23
> > >
> > > However, we appreciate the opportunity to compare our NAR model with L2R. Although L2R cannot be fully reproduced due to the lack of complete implementation details, we still attempted to provide a fair approximate comparison wherever possible to resolve reviewer's concern. Using the publicly available architectural descriptions and parameter specifications from the L2R partially released code, we estimated and compared the **FLOPs of the two models**. (FLOPs measure the number of floating-point operations required by a model and serve as a standard proxy for computational cost and expected runtime.) This comparison offers a reasonable **indication of the relative efficiency** between the two methods on a given instance, in the absence of full reproducibility. We conduct comparisons on TSP across instances with 500, 1000, 3000, 5000, 8000, and 10,000 nodes. We choose TSP rather than CVRP because, in TSP, the AR model’s forward pass naturally scales with the instance size and thus enables more precise prediction. As shown in **Table c**, which reports the FLOPs required to solve a single instance, DEG requires fewer FLOPs than L2R, suggesting that DEG may offer **more efficient and potentially faster inference** under comparable settings. Regarding performance, the limited and non-reproducible nature of the existing L2R implementation prevents us from conducting a reliable comparison.
> > >
> > > We also sincerely thank the reviewer for pointing out this baseline, this is indeed a valuable piece of work, and we have **included it in the AR portion of our related work section**.
> > >
> > >
> > > ##### Table c. FLOPs Comparison between L2R and DEG on TSP
> > >
> > > | FLOPS(M FLOPs) | 500       | 1000      | 3000       | 5000        | 8000        | 10000       |
> > > | -------------- | --------- | --------- | ---------- | ----------- | ----------- | ----------- |
> > > | L2R[1]         | 22.03     | 68.49     | 505.59     | 1340.60     | 3340.00     | 5170.80     |
> > > | DEG            | **13.12** | **49.34** | **425.22** | **1170.10** | **2951.65** | **4650.94** |
> > >
> > > [1] Learning to Reduce Search Space for Generalizable Neural Routing Solver. arXiv, 2025.
> > > #### **W4. Minor Error**
> > >
> > > We thank the reviewer for pointing out the missing citation. We have added the reference to the original diffusion model work (Sohl-Dickstein et al., 2015) at line 60 in the revised manuscript.
> > >
> > > We appreciate the reviewer’s careful reading and valuable feedback. Your comments were particularly insightful and have improved the clarity and completeness of our manuscript. Please feel free to let us know if there are any further issues or suggestions, we would be grateful for any additional guidance.

---

### Author Response · Authors · 2025-11-23
**Global Response**

We thank all reviewers for their thoughtful and encouraging feedback. Reviewers 9NRj and QC6P noted that introducing diffusion-style noise into GFlowNet training is a novel and interesting idea that effectively enhances exploration in high-reward regions. Reviewers QC6P, bkKu, and TAwx highlighted DEG’s strong empirical performance on large-scale TSP and CVRP, with QC6P emphasizing the impressive results on CVRP with N≥5000N. Reviewer bkKu found the methodological components, edge-level reward estimation, diffusion corruption, and TB integration, are clearly motivated, and noted that DEG provides a more general, reward-driven formulation than prior supervised diffusion approaches. Reviewer TAwx agreed that combining GFlowNets with diffusion-based exploration is technically sound and well supported by ablations.

We sincerely appreciate the reviewers’ constructive suggestions, which have greatly improved the clarity, completeness, and rigor of our work. Their comments helped us refine the theoretical explanation, strengthen empirical validation, and clarify the methodological design and experimental protocols. In response to these valuable insights, we have carefully provided detailed answers to each point, revised the manuscript and upload. Below, we provide a global summary of the rebuttal.
* Empirical Evidence of Training Machenism
> Section A.1 and Figure 2 of Appendix in the Paper
* Diversity Experiment
> Table a. Diversity Comparison
* Samll Scale Instance Test
>Table d.1-d.2. Comparison on TSP and CVRP instances with 100 and 200 nodes
* Applicability to Other Combinatorial Optimization Problems
> Table e.1-e.2 Results on SMTWTP and BPP
* Assessment of Learning Efficiency
> Table f.1-f.2 Comparison of Training Efficiency on TSP and CVRP
* Multi-Scale Training
> Table g.1-g.4. Training on Different Scales for TSP/ CVRP and Testing without GSA. with GSA

* GSA Applicability
> Table i.1-i.2. GSA Application to GFACS and AGFN on TSP and CVRP

---

### Meta-Review · Area_Chair_rcPL · 2026-01-06

**Summary:**

The authors presented Diffusion-Enhanced GFlowNet (DEG) framework for solving vehicle routing problems (VRPs), following the previous GFlowNet implementations for VRPs. DEG introduces diffusion-based backward policies when training GFlowNet based on the trajectory balance objective, which is claimed to improve policy diversity for better initial exploration to achieve better performance. A graph-scale adaptor (GSA) was also introduced for generalizable training in DEG with mismatched training and testing scales. The presented results from the original submission and the rebuttal on CVRP, TSP, SMTWTP and BPP demonstrated the effectiveness of DEG and GSA.

**Reviewer Concerns:**

Reviewers 9NRj and TAwx asked for rigorous theoretical analysis of DEG (reviewer bkKu also asked for empirical validation of DEG design). The authors provided theoretical analysis as well as empirical demonstration in rebuttal. However, the authors may need to carefully check the derivation in Appendix A. For example, the equation in line 781 is different from the equations defined in the main text line 207 (eq. 5) and 213 (eq. 6). It is not immediately clear how the authors can state in line 782 that $P_B(\tau) \propto x_t$, which may also be inconsistent with eq. 12 (line 272) in the main text. There are also quite a few vague statements that should be rigorously justified in the revision, for example in line 232-235. The authors may need to consider to proofread the revision for correctness and improve presentation.

Edge-level reward design was questioned by reviewers. The authors claimed in the rebuttal that the introduced computational overhead is minimal and it will lead to "cleaner, low-variance estimate", which is not intuitive and may need detailed explanation. Also, based on eq. 4 (line 198), there can be potential computational issues with increasing scales. the authors may want to explain the mitigation strategies.

Reviewers indicated that some results might be 'cherry-picked'. For example, in quite a few tables, not all the competing baselines are included especially in the additional results. Table 3 and the corresponding discussions in line 421-429 should be revised for consistency and better readability. Also, in Tables 13/14, AFGN with GSA appears to be have similar DEG performances in Tables1/2. The authors may need to discuss and justify the diffusion-enhanced GFlowNet training compared to AFGN indeed leads to the claimed performance improvement. For empirical performance, the authors mostly reported final outcomes from single runs. The authors may considering adding experiments with averages over multiple runs, standard deviations, or confidence intervals, to demonstrate robustness and reproducibility.

The authors are encouraged to open-access data, code and evaluation implementations.

**Reviewer Scores:**

Based on the available rebuttal discussions and remaining concerns listed above, the reviewers may not reach the consensus (especially reviewer 9NRj) for acceptance recommendation. The authors may consider carefully revising the current submission for future venues.

---

### Decision · Program_Chairs · 2026-01-26

Reject